# Archaeal lipostratigraphy of the Scotian Slope shallow sediments, Atlantic Canada

Narges Ahangarian[1], Unyime Umoh[1], Natasha MacAdam[2], Adam MacDonald[2], Patricia Granados[3], Jeremy N. Bentley[1], Elish Redshaw[1], Martin G. Fowler[4], Venus Baghalabadi[1,5], G. Todd Ventura[1]

5    [1]Department of Geology, Saint Mary's University, 923 Robie Street, Halifax, Nova Scotia B3H 3C3, Canada
[2]Nova Scotia Department of Energy, 1690 Hollis St., Halifax, Nova Scotia B3J 3J9, Canada
[3]Centre for Environmental Analysis and Remediation, Saint Mary's University, 923 Robie Street, Halifax, Nova Scotia B3H 3C3, Canada
[4]Applied Petroleum Technology (Canada) Ltd., Calgary, AB T3A 2M3, Canada
10   [5]Department of Pharmacology, Dalhousie University, 5850 College St, Halifax, Nova Scotia, B3H 4R2, Canada

*Correspondence to*: Narges Ahangarian (Narges.ah@hotmail.com), G. Todd Ventura (todd.ventura@smu.ca)

**Abstract.** The Scotian Slope in the North Atlantic Ocean extends for ~500 km along the coast of Nova Scotia, Canada. Its surface sediments host microbial communities, which respond to complex geochemical drivers that not only include communication with the overlying water column, but also potential advection from deeper basinal fluids. Archaea are fundamental components of these communities, and their lipids act as important historical indicators of environmental geochemical change and microbial interactions within marine sediments. This study evaluates the spatial abundance and diversity of archaeal lipids preserved in shallow Scotian Slope sediments to better understand processes. Seventy-four sediment samples from 32 gravity and piston cores, reaching a maximum of 9 meters below seafloor (mbsf) were collected during three survey cruises. In total, 14 archaeal lipid classes comprising 42 unique compounds were detected. The lipid distributions reflect a high contribution of anaerobic methanotrophic (ANME) archaeal groups, such as ANME-1 to -3. Hierarchical cluster analysis and principal components analysis were used to show varying contributions of four lipid classes that included distinct assemblages of intact polar lipids (IPLs), core lipids (CLs), and their degradation products (CL-DPs). IPL to CL and CL to CL-DP turnover rates were estimated for the various lipid classes. Four stratigraphically distinct archaeal lipidomes were observed. The first, reflects a unique community influenced by a nearby cold seep. Three additional ambient sediment lipidomes were detected with overlapping depth intervals. These lipidomes contained varying abundances of IPL, CL, and CL-DPs, which likely mark geochemically controlled, microbial community variations that are further accompanied by a systematic increase to the stockpile of diagenetically altered lipids. The ambient sediment lipidomes appear to be highly spatially conserved across the latitudinal extent of the study area marking a resolvable shallow sediment lipostratigraphy that occupies a sediment stratigraphy that spans ~27,000 ±4,000 years of basin evolution for the Scotian Slope.

# 1 Introduction

The Scotian Slope marks a portion of the North American continental shelf that extends for ~500 km along the eastern coastal seaboard of Nova Scotia (Fig. 1). The slope descends from ~400 m at the shelf edge to ~5000 m water depth near the abyssal plain. Underlying this is a depositional history that began in the Late Triassic leading to 250 million years of continuous sedimentation (Wade and MacLean 1990). Large portions of the deeper sediment basin are affected by salt tectonism, which has greatly impacted basin stratigraphy and locally facilitated hydrocarbon seepage to the ocean seafloor. Deep basin salt tectonic movement has caused localized changes to the sub-basin stratigraphy at all depths resulting the formation of minibasins and in the breaking of petroleum reservoir seals (i.e., Deptuck and Kendall, 2017; 2020) that have in some cases produced hydrocarbon seeps on the ocean floor (Campbell, 2019; Fowler, 2017; Bennett and Desiage, 2022; Chowdhury et al., 2024). How near surface sedimentary microbial ecosystems respond to their geochemical conditions especially when impacted by hydrocarbon seepage is not fully resolved.

The Scotian Slope is therefore an ideal region for examining the interplay between organic matter preservation and microbial interactions, both of which are critical to global carbon cycling. Genomic analyses, including 16S rRNA amplicon sequencing and metagenomic profiling, have resolved diverse microbial communities inhabiting surface and shallow sediments. Bacteria including Proteobacteria, Desulfobacterota, and Caldatribacteriota are the most abundant phyla across various strata (Zorz et al., 2023). Prominent lineages such as Gammaproteobacteria, Deltaproteobacteria, and Alphaproteobacteria are detected in the near surface with Atribacteria, Chloroflexi, and Deltaproteobacteria dominating deeper sediments (Dong et al., 2020). Of the archaeal domain, members of Thaumarchaeota (more recently reclassified as Nitrososphaerota, Rinke et al., 2021) dominate the shallow and surface sediments with decreasing abundance in deeper strata, while phyla such as Methanomicrobia and Lokiarchaeota mark the dominant taxa in deeper buried sediments (Dong et al., 2020). Archaeal communities such as ANME-1 and ANME-2, which play important roles in the anaerobic oxidation of methane, are prominent at depths corresponding to the sulfate-methane transition zone (Dong et al., 2020). The metabolic specificity of these organisms alongside their community composition can be strong indicators of hydrocarbon seepage at targeted Scotian Slope sites (Dong et al., 2020;-Li et al., 2023). For example, microbial communities capable of oxidizing C2+ alkanes dominate the shallow sediments of these hydrocarbon seeps (Zorz et al., 2023). Additionally, microbial communities in sediments close to cold seeps can be enriched in thermophilic bacterial endospores, which produces an influx of deeper biosphere microfauna into the shallow marine sedimentary environment (Gittings et al., 2022; Rattray et al., 2023). As such, the complex microbial communities inhabiting these sediments are characterized as being depth stratified and further influenced by hydrocarbon seepage (Li et al., 2023).

Deep marine sediments are globally extensive, complex, and dynamic biogeochemical interfaces, which serve as sinks for carbon and nutrients (Burdige, 2007). The interaction between microbial communities and sedimentary processes within these environments is a key component of nutrient cycling, organic matter degradation, and environmental changes over geological timescales (Orcutt et al., 2011). A major microbial driver of these processes are archaea, which are ubiquitous in deep marine sediments (Sturt et al., 2004; Fredricks and Hinrichs, 2007; Lipp et al., 2008; Hoshino et al., 2020; Lipp and Hinrichs, 2009;

Biddle et al., 2006). Archaea play critical roles in the transformation of carbon and nitrogen (Offre et al., 2013). Their ability to adapt to diverse and extreme conditions makes them important subjects in the study of life's extremes and global ecological processes (Valentine, 2007; Sollich et al., 2017).

The unique membrane lipid structures of archaea differ from bacteria and eukarya (e.g., Koga et al., 1993; De Rosa, 1996). Archaeal membrane lipids consist of isoprenoidal hydrocarbon chain to the glycerol backbone through ether bonds (De Rosa, 1996). Intact polar lipids (IPLs) such as phospholipids, glycolipids, and ornithine lipids with polar headgroups that can potentially indicate active, or recently active, microbial cells (Sturt et al., 2004; Lipp and Hinrichs, 2009; Schouten et al., 2010; Schouten et al., 2013). In contrast, core lipids (CLs) can be degraded remnants of IPLs where they are then informative of past microbial community dynamics (White et al., 1979) or are the intermediate products of lipid biosynthesis in living cells (Liu et al., 2012a; Meador et al., 2014; Villanueva et al., 2014). Therefore, variation between CLs and IPLs can indicate shifts in microbial community structure and function with sediment depth by diagenesis (Biddle et al., 2006; Schouten et al., 2013).

The composition and distribution of lipid biomarkers, particularly IPLs and CLs, are important in studying marine sediments for tracing microbial life and environmental conditions (Koelmel et al., 2020). The unique compounds such as glycerol dialkyl glycerol tetraethers (GDGTs) can serve as indicators of archaeal activity and in reconstructing past environmental conditions, in terms of the strength of ammonia oxidation, changing sea surface temperature, methane cycling, and pH variations of soils (e.g., Schouten et al., 2002; Kim et al., 2010; Zhang et al., 2011; Hurley et al., 2016; Xiao et al., 2016; Guan et al., 2016).

For this study, the diversity and abundance of archaeal lipids extracted from shallow ocean floor sediments of the Scotian Slope are examined to provide further insights into the microbial processes of deep marine sedimentary systems. The resolvable lipidomes are examined across three spatial dimensions: sediment depth, distance down the continental slope, and along ~3° latitude change of the northwestern trend of the continental margin. They are then compared with several sedimentological, geochemical, and archaeal lipid proxy ratios to better constrain the microbial community structure and function within the sediments as well as to improve constraints on their diagenetic alteration rates.

## 2 Materials and methods

### 2.1 Sample collection

Seventy-four sediment samples were selected from 32 piston and gravity cores collected during three survey cruises that took place aboard the CCGS *Hudson* research vessel in 2015, 2016, and 2018 (Fig. 1 and Table S1). The total area covered was ~40,000 km$^2$, marking ~3° of latitudinal, and ~1500 to 3500 m variation to the ocean depth. Of these, 17 sediment samples were acquired from 12 piston cores during the 2015 expedition. Twenty-four sediment samples were taken from 16 piston cores during the 2016 cruise and 33 sediment samples were obtained from four gravity cores during the 2018 cruise. A 10 m piston corer was used for the 2015 and 2016 cruises (Campbell and MacDonald 2016; Campbell 2019). The 2018 cruise used a gravity corer that extended to 6 m in length. All cores were immediately sectioned into 1.5 m long intervals on-board the ship and inspected for diagnostic signs of hydrocarbon seepage (e.g., gas cracks, bubbling, or strong odours; Campbell and

Normandeau 2019). Sediment within 20 cm of the base of each core was scooped into a 500 mL IsoJar (Isotech Laboratories, Inc.), flushed with nitrogen, sealed, and stored at -20°C for hydrocarbon biomarker and headspace gas analyses (Fowler et al., 2017). The remaining full-length cores were then split longitudinally with a core splitter and further inspected for lithology, obvious hydrocarbon staining, and evidence of gas. Samples for geochemical analyses were wrapped in aluminium foil, sealed

in Whirl-pack® bags, and frozen at -80°C on board the ship. These samples were kept frozen until analysed at the land-based laboratories. Information regarding the specific sampling sites is available in cruise reports (Campbell and MacDonald 2016; Campbell 2019; and Campbell and Normandeau 2019). Among these samples, core "2018, 0007" (Table S1) collected from a gravity core that targeted a suspected cold seep site (Campbell and Normandeau 2019) appears to have been impacted by hydrocarbon seepage based on additional geochemical evidence (Fowler et al., 2018). Several other cores had sediment

intervals potentially indicating the presence of hydrocarbons (Table S2) that were far away from the sampled intervals investigated in this study. Lithologic information was extracted from the open file cruise report Jenner et al. (2022) for core samples collected on 2015 expedition cruise. Sediment lithologic information for the other samples was provided by GSC Atlantic, NRCAN for cores collected during the 2016 and 2018 expedition cruises and are recorded in supplemental Table S2.

## 2.2 Bulk sediment geochemistry

Approximately 10 g of the frozen sediment was weighed, desiccated in a drying oven at 30°C, and reweighed to obtain its dry weight. The dried sediment was powdered using a mortar and pestle and acid digested in 6N $H_3PO_4$ solution for 4 days. Once the inorganic carbon was removed, the sediment was flushed with Milli-Q water, transformed into a slurry using a vortex mixer, and centrifugated at 1250 rpm for 7 min. The supernatant was decanted and the process was repeated until the sediment was renormalized. Decarbonated sediments were then subsampled for bulk geochemical analysis to constrain organic matter

source and diagenetic changes with sediment depth. Sediment total organic carbon (TOC) and total nitrogen (TN) was measured using a Perkin-Elmer 2400 Series II CHNS/O Elemental Analyzer (EA) located in the Centre for Environmental Assessment and Remediation at Saint Mary's University. Stable carbon isotope of TOC ($\delta^{13}C_{TOC}$) measurements were obtained from the University of Calgary.


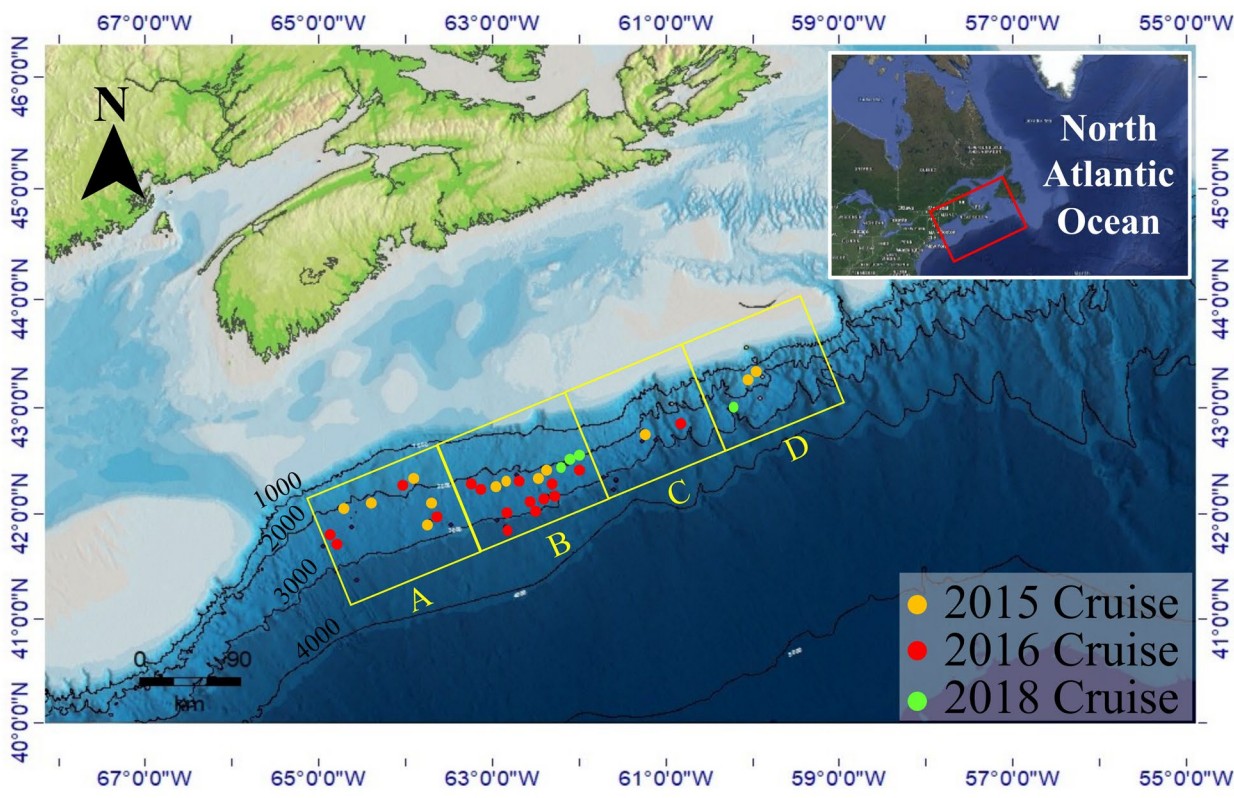

**Figure 1: ArcGIS bathymetric map of the North Atlantic Scotian Margin displaying piston and gravity core locations used in this study. Core locations are grouped into four equal area quadrants (labelled A–D) that extend parallel along the Scotian Slope. Colour circles mark the year that the survey cruise was conducted.**


## 2.3 Porewater Geochemistry

Porewater was extracted from sediments using Rhizon samplers and by centrifugation at 2500 rpm for 10 min. The porewater was decanted into a pre-combusted glass beaker, then filtered through a 0.45-μm filter to remove sediment particles. The exact volume of extracted porewater was recorded as a measure of sediment porosity. Porewater anion concentrations were measured

using a Thermo Scientific Dionex Aquion Ion Chromatography Conductivity Detector System with an anion-exchange column and a DS6 Heated Conductivity Cell fitted with an AERS_4mm suppressor pump and a Dionex AXP Auxiliary pump and pump ECD from the Saint Mary's University Organic Geochemistry Laboratory (OGL). The IC was further configured with an in-line Thermo Scientific 9×24mm Dionex InGuard Ag sample prep cartridge and Thermo Scientific Dionex InGuard Na prep cartridge to facilitate trace analysis of seawater. The system was controlled via Thermo Scientific Chromeleon 7

chromatography system version 7.3 software. Seven Anion Standard II (from Sunnyvale, California) in deionized water. The anion standard (S+D) was an amalgamation of $H_2O$ (99.9%, CAS# 7732-18-5) and the following anions – $F^-$ (20 mg $L^{-1}$, CAS#

7681-49-4); $Cl^-$ (100 mg $L^{-1}$, CAS# 100 mg $L^{-1}$); $NO_2^-$ (100 mg $L^{-1}$, CAS# 7632-00-0); $Br^-$ (100 mg $L^{-1}$, CAS# 7647-15-6); $NO_3^-$ (100 mg $L^{-1}$, CAS# 7631-99-4); $PO_4^{3-}$ (200 mg $L^{-1}$, CAS# 7778-77-0); $SO_4^{2-}$ (100 mg $L^{-1}$, CAS# 7757-82-6), stored in -4 °C refrigerator. Carbonate and Sulfite are prepared as separate stock solutions. Carbonate is prepared using Anhydrous

Sodium Carbonate ACS powder from Fisher Chemicals (CAS # 497-19-8) and $SO_3^{2-}$ is prepared using Anhydrous Sodium Sulfite crystalline powder from Fisher Chemicals (CAS# 7757-83-7). A seven-anion standard mixture was diluted to 0.5, 1, 2, 5, 10, 20, and 50 ppm to generate an external calibration curve. The high salinity of marine porewater requires that the system be equipped with two in-line guard cartridges to remove chloride ($Cl^-$) and sodium ($Na^+$) ions from the sample before reaching the anion-exchange column.

## 2.4 Lipid extraction

Before extraction, PAF (1-alkyl-2-acetyl-sn-glycero-3-phosphocholine) was added as a recovery standard into the samples. Total lipid extracts (TLE) were recovered with a modified Bligh and Dyer extraction technique (Bligh and Dyer, 1959; Sturt et al., 2004) as described in Bentley et al. (2021). The liquid-liquid extraction method employed a blend of polar and non-polar solvents that effectively separates the organic phase from the inorganic phase, while gently lysing cellular membranes.

## 2.5 Lipid separation and identification

Aliquots comprising 1 % and 3 % of the TLE were injected into an Agilent Technologies 1290 Infinity II ultra-high performance liquid chromatograph (UHPLC) coupled to a 6530-quadrupole time-of-flight mass spectrometer (qToF-MS) run in reverse phase using electrospray ionization. Liquid chromatographic separation used a ZORBAX RRHD Eclipse Plus C18 (2.1-mm×150-mm×1.8-μm) reverse phase column, equipped with a guard column, and maintained at a stable temperature of

45 °C throughout the sample analysis. The injection solvent was methanol. The mobile phase flow rate was set at 0.25 mL $min^{-1}$ with the composition of mobile phases being methanol/formic acid/ammonium hydroxide ([100:0.04:0.10] v:v:v) for mobile phase A, and propan-2-ol/formic acid/ammonium hydroxide ([100:0.04:0.10] v:v:v) for mobile phase B. The mobile phase composition began with 100 % A for 10 min., stepped to a linear addition of B to 24 % held for 5 min., followed by a linear gradient to 65 % B for 75 min., then 70 % B for 15 min., that completed by re-equilibrating with 100 % A for 15 min.

Compounds were tentatively identified by mass spectral analysis in conjunction with expected chromatographic elution patterns (Table S3) as found in the literature (e.g. Wörmer et al., 2013; Wörmer et al., 2015; Yoshinaga et al., 2011; Schouten et al., 2008; Schubotz et al., 2009; Liu et al., 2012a). Lipid quantification used $C_{21}$-PC (1, 2-diheneicosanoyl-*sn*-glycero-3-phosphocholine) as an internal standard and was based on accurate mass detection of $[M^·+H]^+$, $[M^·+NH_4]^+$, and $[M^·+Na]^+$ adducts using Agilent Technology Mass Hunter software. Lipids concentrations were then normalized to extracted sediment

mass (μg g $sed^{-1}$) and sample TOC mass (μg wt.% $TOC^{-1}$).

## 2.6 Calculation of TOC decay rate for lipid normalization

Organic matter systematically decreases by diagenetic processes as sediments become more deeply buried. These changes result in a depth bias for lipid concentrations normalized to sediment TOC. To correct for this, the inferred TOC value at depth $i$ marking the original point in time of sediment deposition ($TOC_{adjusted_i}$) was calculated as the sum of the measured TOC content in the sediment samples ($TOC_{measured_i}$) plus the TOC that is predicted to have decayed ($TOC_{decayed_i}$) following Eq. 1:

$$TOC_{adjusted_i} = TOC_{measured_i} + TOC_{decayed_i}, \tag{1}$$

The decayed TOC was inferred based on a regionalized Scotian Slope sediment TOC depth profile that extended to 8.71 mbsf as calculated from the 74 samples collected from this study as well as 11 additional shallow sediment samples (<1 m) from other ambient sediments. The resulting diagenetic decay curve (Fig. 2 and Eq. 2)

$$y = -366.2 \ln (TOC) + 47.102, \tag{2}$$

was then used to predict a regionalized reference TOC ($TOC_{predicted}$) value for any depth (y) within the study (Eq. 3).

$$TOC_{predicted} = e^{\frac{47.102-y}{366.2}}, \tag{3}$$

The TOC decay rate ($n$) varies with sediment depth and must be determined specifically for each 1 m burial depth interval (y) following Eq. 4:

$$f(n_i) = \frac{d(TOC_{predicted_i})}{dy_i} = -\frac{TOC_{predicted_i}}{366.2}, \tag{4}$$

$$TOC_{decayed_i} = \sum_1^i (n_i \times TOC_{predicted_{i-1}}), \tag{5}$$

where $n_i$ is the TOC decay rate at depth $i$. The $TOC_{decayed}$ and $TOC_{adjusted}$ were then calculated at each depth using Eq. 5 and Eq. 1 respectively and following the modelled TOC decay rate in Eq. 5 Table S4, and Fig. S1.

## 2.7 Statistical analysis

A hierarchical cluster analysis (HCA), heatmap dendrogram of the sediment sample lipids, was generated with the R statistical software environment (R Core Team, 2023) using the "heatmap" function of the "gplots" package. Lipid abundances were transformed to their respective z-scores that then served as the data matrix. *Ward's* minimum variance unbounded distance function was applied to calculate the dissimilarity of the data matrix. Matrix cells were coloured according to a gradient scale representing the z-scored values. Minitab statistical software was then used to perform principal components analysis (PCA). PCA was performed on the lipid concentration data, which was transformed to a correlation matrix using a Mahalanobis distance, that was then used to calculate the centroid from the covariance structure of the data.

## 3 Results

### 3.1 Bulk sedimentary organic matter and porewater geochemistry

Sedimentary organic carbon ranged from 0.12 to 1.1 wt. % with an average of 0.5 wt. % (Fig. 2 and Table S1). This was calculated based on 85 TOC measurements produced from 33 sediment cores. A sharp down core decrease in TOC is observed from 0 to 0.5 mbsf, which is followed by a more gradual decrease with core depth. The TN values ranged from 0 to 0.12 wt. % with an average of 0.07 wt. % (Fig. 2 and Table S1). Samples "2016-049, 443-448 cm" and "2015-029, 490-495 cm" comprise the lowest and highest TN values, respectively. The $\delta^{13}C_{TOC}$ values range from -21.7 to -24.4 ‰ (Fig. 2 and Table S1), with one outlier (-27.4 ‰ collected from seep site core "2018, 0007"). Collectively, these trends are consistent with shallow marine sediments experiencing early diagenetic alteration of its organic matter. The trends also indicate that while mass transport deposits likely have occurred within the upper 9 m of the sampled sediments, the impact has not changed the average organic matter loading profile of the slope.

Porewater anion measurements were conducted from 7 gravity and piston cores. The porewater anion concentrations were measured to capture carbon, nitrogen, and sulfur biogeochemical cycling. Low nitrate concentrations (~0.1 to ~0.2 mmol L$^{-1}$) steadily decrease over the 8m sediment depth (Fig. 2). Even lower nitrite concentrations (~0.05 to ~0.14 mmol L$^{-1}$) show an antiparallel trend and slightly increase with sediment depth. Dissolved inorganic carbon (DIC) steadily increased with depth (from near zero to ~8 mmol L$^{-1}$). An inverse sulfate depletion trend occurs with sediment depth (~28 mmol L$^{-1}$ to ~10 mmol L$^{-1}$). Together these sulfate and DIC mark the most dominant porewater geochemical feature that enables the identification of the sulfate alkalinity transition zone (SATZ). For the Scotian Slope, the SATZ resides at ~3.25 mbsf. This zone stratigraphically sits above the sulfate methane transition, which was not measured in this study.

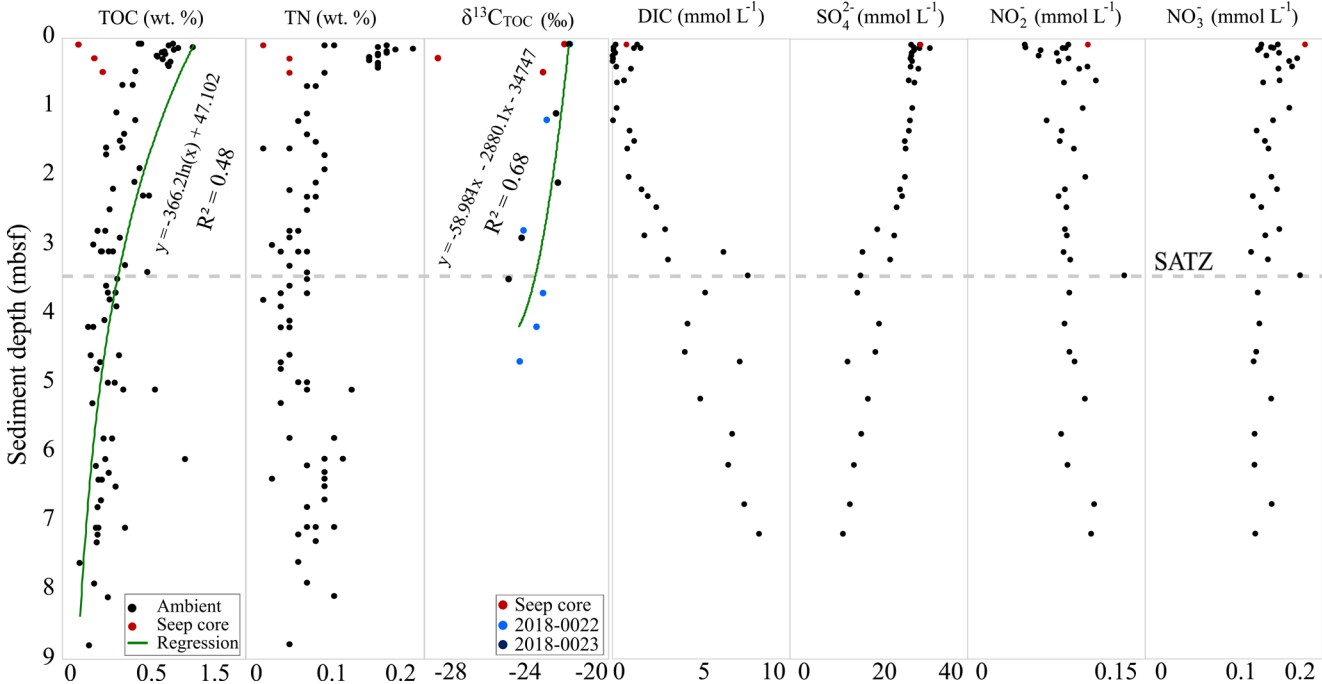

**Figure 2: Down core TOC, TN, DIC, pore water anions, and stable carbon isotope ($\delta^{13}C_{TOC}$) trends of the Scotian Slope sediment samples. Dotted line indicates the SATZ.**

### 3.2 Diversity of archaeal lipids in the Scotian Slope sediments

A total of 14 archaeal lipid classes comprising 42 unique compounds were tentatively identified and quantified across 74 sediment samples using mass spectrometric techniques and comparisons of elution times from the literature (Fig. 3 and Table S3). Lipid classes were grouped based on the hypothesized degree of preservation into IPLs, CLs, and CL-DPs. Additionally, upper water column plant and algae-based chlorophyll-*a* (Chl-*a*) and hydroxy-chlorophyll-*a* (OH-Chl-*a*) were identified and quantified as outgroup allochthonous additions to the seafloor.

Of these, eight IPL classes comprising 18 unique compounds were identified (Fig. 3 and Table S5). Detected mono-glycosidic GDGTs (1G-GDGTs) comprising 0–3 rings and crenarchaeol (Cren). For this series 1G-GDGT-0 and 1G-Cren were the most abundant compounds. For di-glycosidic-GDGTs (2G-GDGTs), compound classes included 0–2 rings and Cren, with the acyclic lipid being the dominant compound. Detected hydroxylated 1G-GDGTs (1G-OH-GDGTs) were -0 to -2, with 1G-OH-GDGT-0 being the dominant compound. Hydroxylated 2G-GDGTs (2G-OH-GDGTs) ranged from 0–2 with the dominant compound also being 2G-OH-GDGT-0. Apart from monolayer lipid structures, four $C_{40}$ bilayer IPLs were detected. These included mono- and di-glycosidic (1G- and 2G-ARs), phosphatidicacid, (PA-AR), and hydroxyphosphatidicacid (PA-OH-AR) archaeols.

Six classes of CL and CL-DPs comprising 24 unique compounds were identified (Fig. 3 and Table S6). Commonly detected GDGTs were GDGT-0 to GDGT-3, Cren, and the stereoisomer of crenarchaeol (Cren′) (Liu et al., 2018; Sinninghe Damsté et al., 2018). For these compounds, GDGT-0 and Cren were the most abundant archaeal lipids in the surveyed region of the Scotian Slope sediments. Detected hydroxyl glycerol dialkyl glycerol tetraethers (OH-GDGTs) comprised OH-GDGT (0–3, and Cren) with OH-GDGT-0 being the dominant compound were also observed in all surveyed samples. Archaeol was detected in all sediment samples, where it typically marked the third most abundant archaeal lipid on most sediment samples (Fig. 3 and Table S5).

Archaeal CL-DPs take on many forms (e.g., Liu et al., 2016). For this study, only 13 derivatives were targeted. These putatively include the glycerol dialkanol diethers (GDDs) 0–4 and Cren, with GDD-0 and GDD-Cren being the dominant compounds; hydroxylated glycerol dialkanol diethers (OH-GDDs) 0–2, dominated by OH-GDD-0; and biphytanic diols 0–3 (BpDiols) with BpDiol-0 being the most prevalent highly degraded archaeal biomarker (Fig. 3 and Table S6).

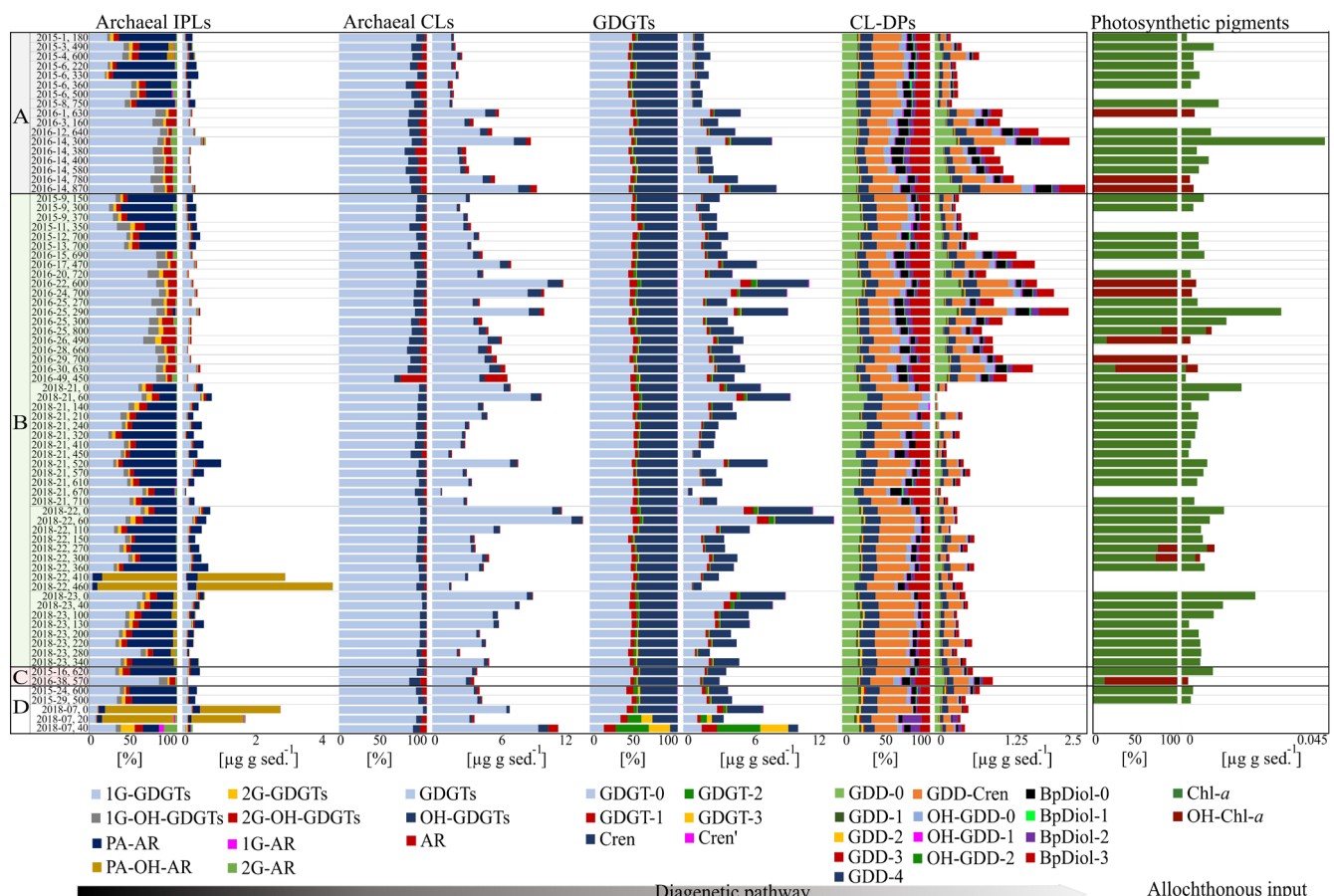

**Figure 3: Relative and absolute abundances of archaeal IPL, CL, CL-DP, and photosynthetic pigments within the four Scotian Slope quadrants (A-D; Fig. 1) core sediments (sample data provided in Table S1).**

# 4 Discussion

## 4.1 Chemotaxonomic relationships

In marine environments, isoprenoidal GDGTs are the main lipids synthesized by ammonia-oxidizing Thaumarchaeota that inhabit both the surface ocean and mesopelagic zone (Sinninghe Damsté et al., 2002; Schouten et al., 2002; Church et al., 2010; Villanueva et al., 2015; Zeng et al., 2019). However, members of Phylum Thaumarchaeota have different compositions of IPL and CL-GDGTs, depending on their habitat (Bale et al., 2019; Elling et al., 2017). IPLs, in addition to taxonomy, could be reflective of various habitats of Thaumarchaeota (Elling et al., 2017).

With respect to individual IPL classes, 1G-GDGTs, 2G-GDGTs, and 2G-OH-GDGTs are abundant in all Thaumarchaeotal strains (Elling et al., 2017, 2015) and have been used for tracing planktonic thaumarchaeal biomass (Elling et al., 2017) in the marine water column (Schubotz et al., 2009; Schouten et al., 2012; Wakeham et al., 2012; Basse et al., 2014; Xie et al., 2014). Additionally, the synthesis of 1G-GDGTs and GDGTs at seep sediments is also attributed to Bathyarchaeia, previously known as the miscellaneous Crenarchaeotal group (MCG) (Zhang et al., 2023) and may be a reflection of the elevated lipid abundances that are found in these settings.

For CLs, OH-GDGTs have been widely detected in marine environments (Liu et al., 2012b; Huguet et al., 2013; Varma et al., 2024a; Elling et al., 2017; Sinninghe Damsté et al., 2012; Liu et al., 2012b). They are absent in Thaumarchaeota group I.1b, but abundant in Thaumarchaeota group I.1a (Schouten et al., 2013). Unspecified Crenarchaeota or Euryarchaeota have been proposed to synthesize OH-GDGTs (Lü et al., 2015). High temperature enhances production of OH-GDGTs with a higher degree of cyclization (Lü et al., 2015; Umoh et al., 2022). Culture experiments indicated organisms have different compositions of GDGTs and OH-GDGTs (Elling et al., 2017; Bale et al., 2019) and IPL OH-GDGTs in their membrane (Elling et al., 2015, 2017), which can be used as taxonomic indicator within strains of phylum Thaumarchaeota. Cren´ has been detected in living archaeal cells such as ''*Ca.* Nitrosotenuis *uzonensis*" indicating it is not a product of diagenesis (Sinninghe Damsté et al., 2018). Higher production could indicate biological adaptation to environmental stresses, such as high temperature, as the membrane of some thermophilic Thaumarchaeota contain high abundances of Cren´ (Pitcher et al., 2010; Sinninghe Damsté et al., 2012). Low contents of GDGT-1 to -3 and Cren´ with relatively high concentration of Cren and GDGT-0 in environmental samples indicate non-thermophilic Thaumarchaeota source of GDGTs (Schouten et al., 2000; Schouten et al., 2013; Schouten et al., 2002). GDGT-0 is also produced by methanogenic Euryarchaeota (Blaga et al., 2009). Archaeol is synthesized by members of Euryarchaeota, Crenarchaeota, and Thaumarchaeota (Koga et al., 1993). In environmental samples, the detection of AR is mainly associated with methanogens (Pancost et al., 2011) and ANME (Rossel et al., 2008).With regards to CL-DPs, putative GDDs are widely detected in marine sediments (Knappy and Keely, 2012) where they are regarded as either membrane components or intermediate structures in GDGT biosynthesis (Meador et al., 2014; Coffinet et al., 20215; de Bar et al., 2019) or degradation products of GDGTs (Lui et al., 2012; Coffinet et al., 2015; Mitrović et al., 2023; Hingley et al., 2024). IPL-based GDDs have been detected in cultures of various methanogenic archaea, such as *Methanosarcina mazei* and *Methanosphaera stadtmanae* (Liu et al., 2012; Meador et al., 2014; Bauersachs et al., 2015), implying that GDDs are also

biosynthesized by living organisms although the exact pathways have not been determined. 1G-GDDs were reported in Ca. *N.*

*maritimus* (Elling et al., 2014; Meador et al., 2014b), and high abundances of both GDDs and 1G-GDDs are reported in *N. viennensis* strains of phylum Thaumarchaeota (Elling et al., 2017).

## 4.2 Regional sediment depth trends

Lipid concentrations from sediment samples were binned into 1 m thick stratigraphic intervals to derive regional down-core

depth trends (Fig. 4A-L). The sediment and TOC normalized concentrations of IPLs, CLs, CL-DPs, and upper water column photosynthetic pigments were then averaged across all samples falling into its specific depth interval (Fig. 4 and Fig. S2). The 14 lipid classes have distinct trends with both increasing and decreasing loadings with sediment depth that are further complicated by occasional lack of consistency between the two normalization schemes. These variations strongly suggest the lipid class are derived from different source inputs.

Resolving archaeal lipid source inputs is difficult and the subject of long-term disagreement (e.g., Zhang et al., 2011; Hu et al., 2015; Li et al., 2016; Guo et al., 2018; Cheng et al., 2021; Umoh et al., 2022). For this study, sourcing considers the following framework. Comparisons between the two lipid normalization methods enable a basic premise for determining allochthonous versus autochthonous input. This is because normalization by sediment TOC adjusts the lipid concentration relative to what is largely an allochthonous input of upper water column supply of detrital sedimentary organic matter.

Alternatively, because the ocean floor sediments contain relatively low organic matter abundances (<1.1 wt. %) the method of normalizing lipid concentrations to sediment volume imparts little influence from upper primary productivity. Therefore, when these two normalization schemes produce similar depth trends, the quantified lipid is likely derived from the overlying water column. However, if dissimilar down core profiles are found the lipid is more likely to have been sourced from within its host sediment. Additionally, euphotic zone produced photosynthetic pigments, Chl-*a* and OH-Chl-*a*, should also provide a

stratigraphic record of changing productivity through time. These pigments alternate between high and low loadings and have near identical depth trends across both normalization methods (Fig. 4L). We therefore interpret other lipid classes as being largely sourced from the upper water column if: 1) their down-core profiles are the same across both normalization schemes and 2) their stratigraphic loadings match the measured Chl-*a* and OH-Chl-*a* depth profiles (Fig. 4L). We further interpret lipid classes as being sourced within the sedimentary environment if sediment normalized depth trends dominate over their

respective TOC normalized counterparts.

Normalized down core concentration profiles suggest 1G-GDGTs and GDGTs -0, -1, and -Cren are largely dominated by upper water column inputs as their two normalized stratigraphic trends largely replicate the photosynthetic pigment profile (Fig. 4A and F). However, an additional input of sedimentary-sourced lipids appears to arise over the 5 to 9 m depth range. 2G-GDGTs appears to be sourced from within the upper 0-3 mbsf of sediments (Fig. 4C) as neither normalization profile

matches Chl-*a* and the mean lipid concentration at 1 m much greater than what is observed for all other depths. 1G-OH-GDGT-0, -1, and -2 appear to be sourced from the water column as their TOC normalized abundances are largely weighted to the

TOC normalization method for these intervals (Fig. 4B). Also similar to their IPL precursors, OH-GDGTs also appear to be sourced from the water column as the compounds have more dominant TOC normalized abundances. IPL-based ARs appear to be largely sourced from sediments as their abundances are distinctly different from that of the photosynthetic pigments down core trends and are dominated by sediment normalized concentration trends (Fig. 4H). For CL-DPs, the larger TOC normalized contributions of GDDs, OH-GDDs, and BpDiols suggests these compound's precursors were largely derived from the water column (Fig. 4 I-K). In summary, water column loading is the main source for 1G-GDGTs in the Slope. 2G-GDGTs, OH-GDGTs, ARs, and CL-DPs have a mixed source, but largely derived from within the ocean floor sediments. As much of the quantified lipid classes arise from the water column and the near surface, they are potentially prone to being diagenetically altered with time as they become more deeply buried.

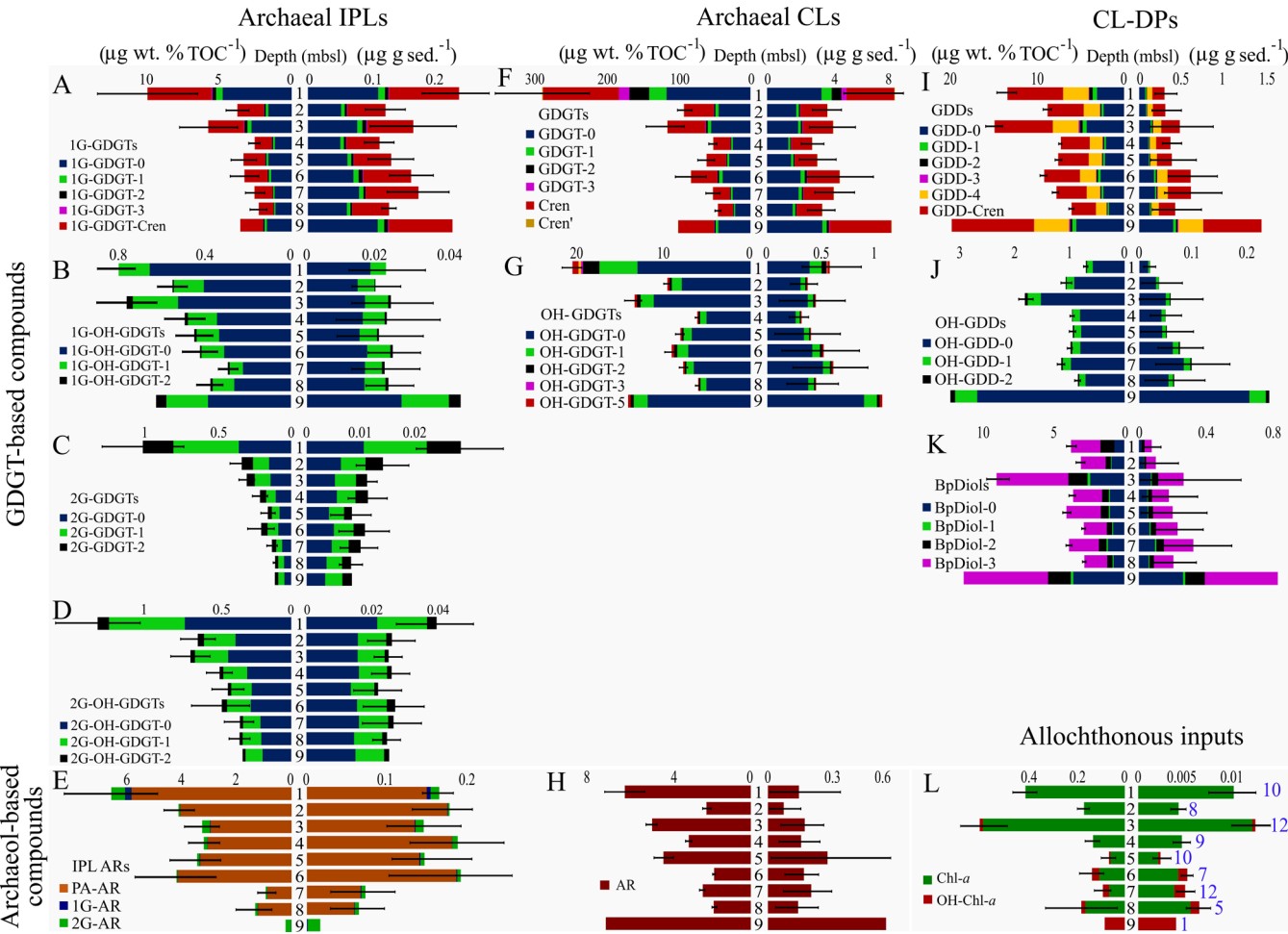

**Figure 4: Bar graphs of down-core archaeal lipid class abundances normalized to sediment (right) and TOC (left) (blue font number in lower right bar graph indicates the summed samples for each depth interval). Black lines through the bars of each graph mark standard deviations of lipid abundance measurements.**

### 4.3 Preservation potential of archaeal lipids

Intact polar membrane lipids, especially IPL-GDGTs, have been used as biomarkers for living microbial communities in marine sediments (Biddle et al., 2006; Lipp et al., 2008; Schubotz et al., 2011; Rossel et al., 2011; Xie et al., 2014; Evans et al., 2017; Carr et al., 2017) under the assumption that these compounds rapidly degrade to CLs upon cell death. Detected IPLs in sediments were therefore considered to be produced by in situ microbial communities. However, degradation models of sedimentary IPLs indicate that there may also be substantially IPL contributions to fossil compounds as these compounds can be preserved across geological time scales (Lipp and Hinrichs, 2009; Schouten et al., 2010; Lengger et al., 2013, 2014a; Lin et al., 2013; Xie et al., 2013; Wu et al., 2019), showing that they are not accurate biomarkers for living microbial biomass.

GDDs are ubiquitously found in marine sediments (Knappy and Keely, 2012) and are generally known as GDGT degradation products (Lui et al., 2012; Coffinet et al., 2015; Mitrović et al., 2023; Hingley et al., 2024). Because of their chemical structure, both sources of GDGT biosynthesis intermediates compounds (Liu et al., 2012a; Meador et al., 2014; Villanueva et al., 2014) and GDGTs degradation products (Knappy and Keely, 2012; Liu et al., 2012a; Yang et al., 2014) have been proposed. However, GDDs have been reported to have similar microbial origin to GDGTs rather than being a direct diagenetic product of them (Mitrović et al., 2023).

A simple diagenetic alteration pathway is further examined to help resolve lipid sourcing relationships (Fig. 5A-D) based on the structural complexity of various lipid classes. Lipid abundances were compared to test whether the model conforms to the conditions measured within the slope. A moderate correlation was found between the occurrence of 1G-GDGTs and GDGTs ($R^2 = 0.48$; Fig. 5A), which could arise from additional lipid incorporation within the sediments (see Section 4.2). It may equally derive from different living archaeal communities that host both IPL and CLs as part of their membrane structures (e.g. Ingalls et al., 2012), as well as a lack of diagenetic alteration owing to the recalcitrant nature of glycosidic headgroups (Lipp and Hinrichs, 2009; Wu et al., 2019; Bentley et al., 2022). Similarly, a moderate correlation also exists between GDGTs and GDDs ($R^2 = 0.59$ Fig. 5B; for the continuous series of lipidomes A3 and A4s that marks a continuous deeper sediment interval; see Section 4.5). GDGT sources do not directly lead to BpDiol production ($R^2 = 0.05$, Fig. S3). Instead, BpDiol appears to be almost exclusively sourced from GDDs ($R^2=0.75$; Fig. 5D).

As several structurally related lipid classes may be genetically related, mean slope degradation proxies (Eq. 6 – 17; Table S7) were calculated and plotted as 1m interval depth profiles (Fig. 6A-E). From these data, 1G-GDGTs and OH-GDGTs are not only observed to be quite resilient within the sedimentary environment; they may also have increased relative IPL concentration across different sediment depth intervals (Fig. 6A). Lipids with 2G headgroups appear to be less stable with a ~0.3 (for 2G-GDGT) and a 0.5% (for 2G-OH-GDGT) per meter headgroup loss rates (Table S8; Fig. 6B). GDGTs are observed to transform to GDDs at a rate of 1.4 % m$^{-1}$ (with a similar 1.6% rate for OH-GDGTs). GDDs appear to systematically give rise to BpDiols (Fig. 6D). Following this general profile, there is an 18.3% turnover in the GDDs pool relative to BpDiols from the surface to the bottom portion of the sediment profile, marking a 2% m$^{-1}$ transformation rate with sediment burial (Table S8). PA-AR,

1G-AR, and 2G-AR also appear to systematically degrade to AR at 7, 0.4, and 0.5% m[-1], respectively. These results indicate the diagenetic alteration of lipids occurs, for most lipid classes following expected degradation paths over the ~27±4 Kyr (Jenner et al., 2022; Table S1; Fig. S4) interval that marks the 9 m surveyed depth of the Scotian Slope. The rate of change differs by the chemical structure with C80 structures having the following stability trends CL-DP (GDD) to CL-DP (BpDiol) >> CL to CL-DP > IPL to CL. C40 lipids follow a slightly different behavior with IPL-AR (PA) to CL (AR) >> IPL-AR (1G & 2G) to CL (AR) (Table S8).

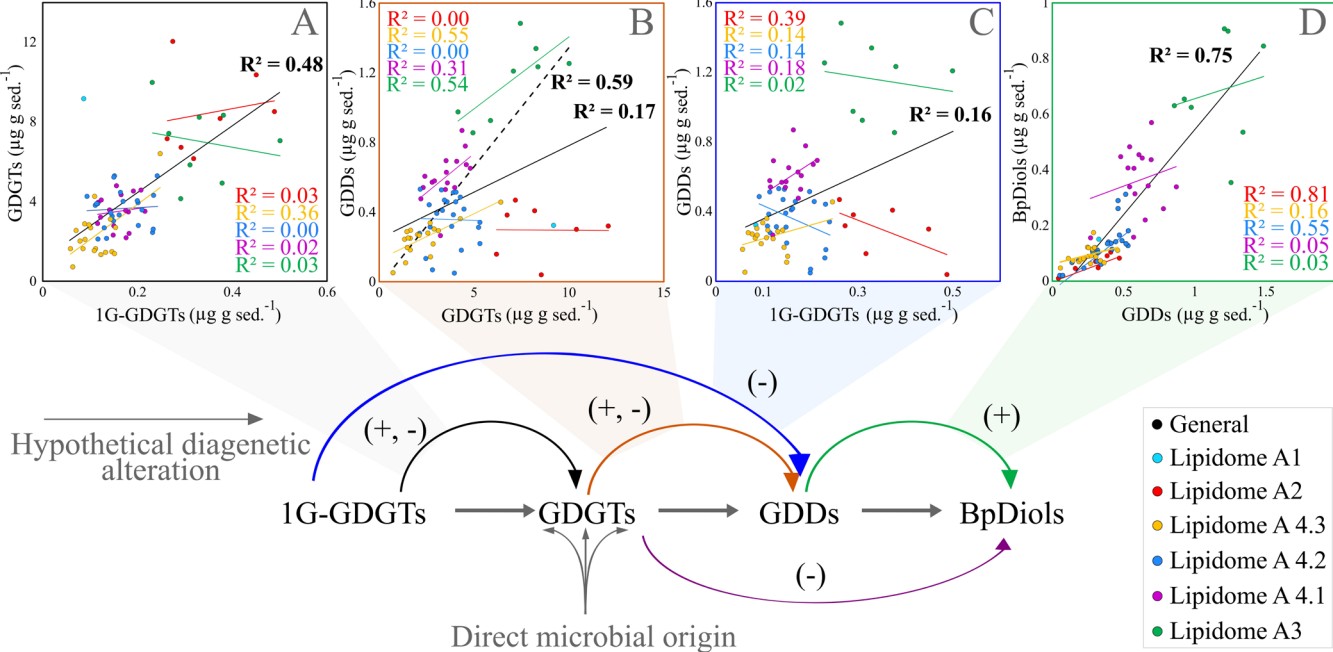

**Figure 5: Cross plots of various lipid class concentrations associated within an expected diagenetic alteration pathway (bottom flow chart). Parentheses with plus and minus signs indicate statistically relevant agreements to the pathway. Lipidomes in the legend are discussed in Section 4.5.**

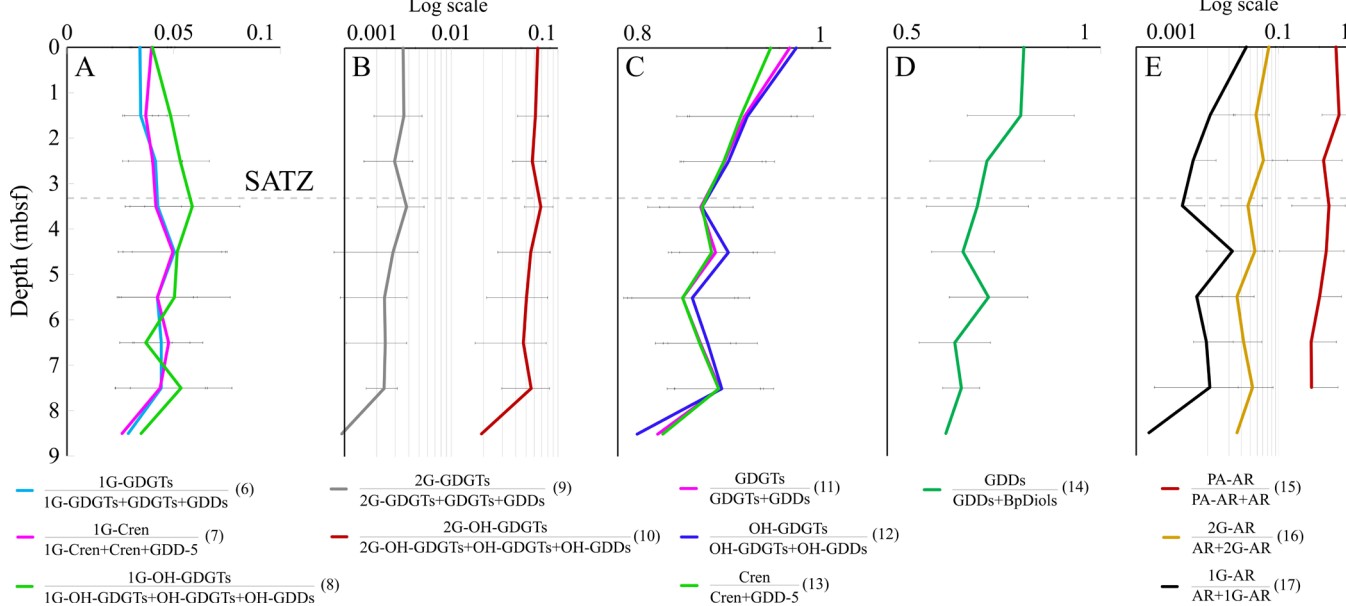


**Figure 6: Lipid precursor to product pair changes with core depth. Dotted line indicates the SATZ. Numbers in parentheses label Eq. 6 – 17.**

## 4.4 Archaeal lipid proxies for methane seepage

Sediments impacted by hydrocarbon seepage typically have unique microbial communities that include ANMEs (i.e., Boetius et al., 2000; Hinrichs et al., 2000; Pancost et al., 2001; Knittel and Boetius, 2009). To investigate this, down-core profiles of three archaeal lipid proxies were calculated for both CLs and IPLs to further elucidate the biogeochemical processes influencing lipid distributions (Fig. 7; Table S7). Of these, the methane index (MI; Eq. 18) (Zhang et al., 2011; Guan et al., 2016) measures the proportion of ANME-1 Euryarchaeota favoured GDGT-1 to -3 within sediments (Pancost et al., 2001;

Blumenberg et al., 2004). MI values spanning 0.3–0.5 indicating a transition between normal marine to a hydrocarbon impacted environments (Zhang et al., 2011). Values >0.5 indicate the presence of active methanotrophic archaeal communities.

$$MI = \frac{[GDGT-1]+[GDGT-2]+[GDGT-3]}{[GDGT-1]+[GDGT-2]+[GDGT-3]+[Cren]+[Cren']}, \qquad (18)$$

For the ambient sediments of the Scotian Slope, MI values were consistently low across all sediment depths (Table S9). Ambient sediment IPL-MI ranged from 0.07 to 0.50 (avg. = 0.15) and do not systematically change with depth, highlighting a

slightly higher proportion of ANME-1 communities within deeper sediments. Core 2018-0007's hydrocarbon seep sediments IPL-MI values range from 0.12 to 0.47; averaged at 0.30 for. Ambient sediment CL-MI have values ranging from 0.08 to 0.27 (avg. = 0.15). Higher CL-MI values are recorded from the seep sediments (ranging from 0.22 to 0.90; avg. = 0.56) suggesting the presence of methanotrophic archaea (Zhang et al., 2011; Fig. 7).

The ratios GDGT-2/Cren (Zhang et al., 2016) and GDGT-0/Cren (Blaga et al., 2009) characterize methanogenic archaeal

contribution to GDGT pool. GDGT-0/Cren and GDGT-2/Cren ratios >0.2, along with MI >0.3, indicate methane cycling

(Weijers et al., 2011; Zhang et al., 2016; Umoh et al., 2022). For the Scotian Slope, GDGT-0/Cren ratios were low ranging

from 0.84 to 1.86. GDGT-2/Cren ratios ranged from 0.02 to 4.37 (avg. 0.05 for ambient sediments). Collectively these ratios

indicate non-methanogenic contributions to the archaeal pool for the ambient slope sediments. Considerably higher values

(avg. 1.34 and 1.67) consistent with active methane cycling (e.g., Teske et al., 2018) were observed within the seep sediments.

With the exception of seep impacted sediments, all biomarker proxies indicate little activity with respect to methane cycling.

No systematic changes to ratio values are coupled to any measured porewater geochemical trends (Fig. 2).

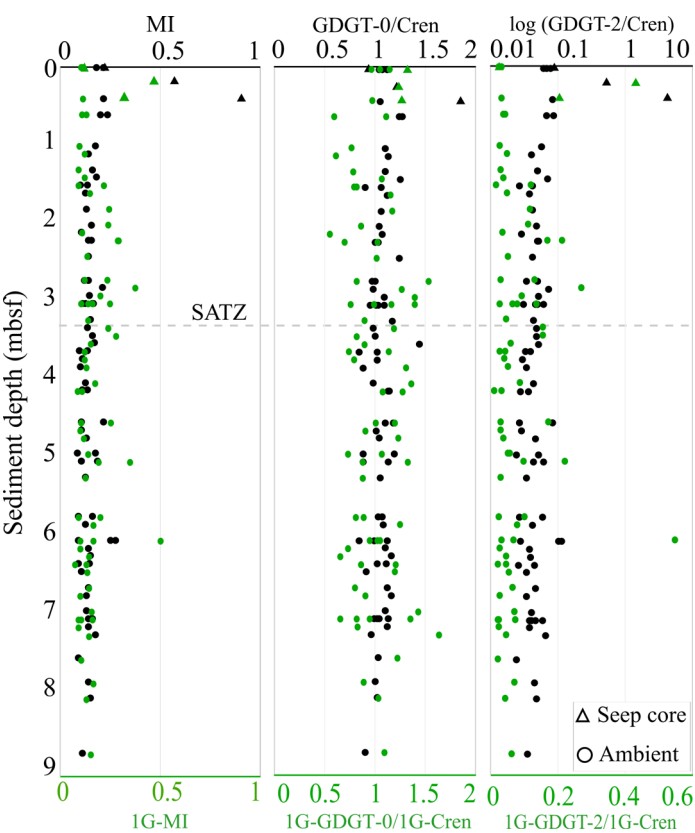

**Figure 7: Down core profiles of MI, GDGT-0/Cren, and GDGT-2/Cren lipid ratios for CLs (labelled in black) and 1G-IPLs (labelled in green). Dotted line indicates the SATZ.**

### 4.5 Scotian Slope archaeal lipidomes

### 4.5.1 Resolution of lipidomes

A heatmap dendrogram of sediment normalized lipid concentrations was calculated to resolve slope-wide diversity patterns (Fig. 8). From this, four distinct groups of lipid classes (labelled as 1a, 1b, 1c, 2, 3, and 4) were resolved. These classes mostly arise from distinct assemblages of IPL, CL, CL-DPs, but also mark differences in water column (i.e., lipid class1a) and sediment (i.e., lipid classes 1b, 1c, 2, and 3) inputs. Additionally, six groups of samples including four main groups and 3 sub-groups (labelled as lipidomes A1, A2, A3, A4.1, A4.2, and A4.3) were further identified representing complex assemblages

of living, fossil, and further degraded lipids within the slope sediments.

PCA was performed to further validate the results of the heatmap dendrogram. The results are displayed as both factor loading and score plots with components extracted using eigenvalues greater than one. The first two principal components, accounting for 85 % of the variance in the data, were selected for visualization (Fig. S5). The factor loading plot (Fig. 9) yields three primary lipid clusters (labelled 1–3), with further subdivisions (labelled 1a, 1b, 2, 3a, and 3b). The score plot shows sample

clusters that largely group by sediment depth (Fig. 9). Additionally, the scores plot with samples grouped by their location within the four quadrants of the Scotian Slope, as outlined in Figure 1, indicate no preferential lipidomic changes along the length of the slope (Fig. S6). Collectively, these data largely mirror the patterns observed in the heatmap dendrogram (Fig. 8). The agreement between the two statistical measures suggests specific lipidomic signatures are characteristic features of the deep marine sediments across the Scotian Slope.


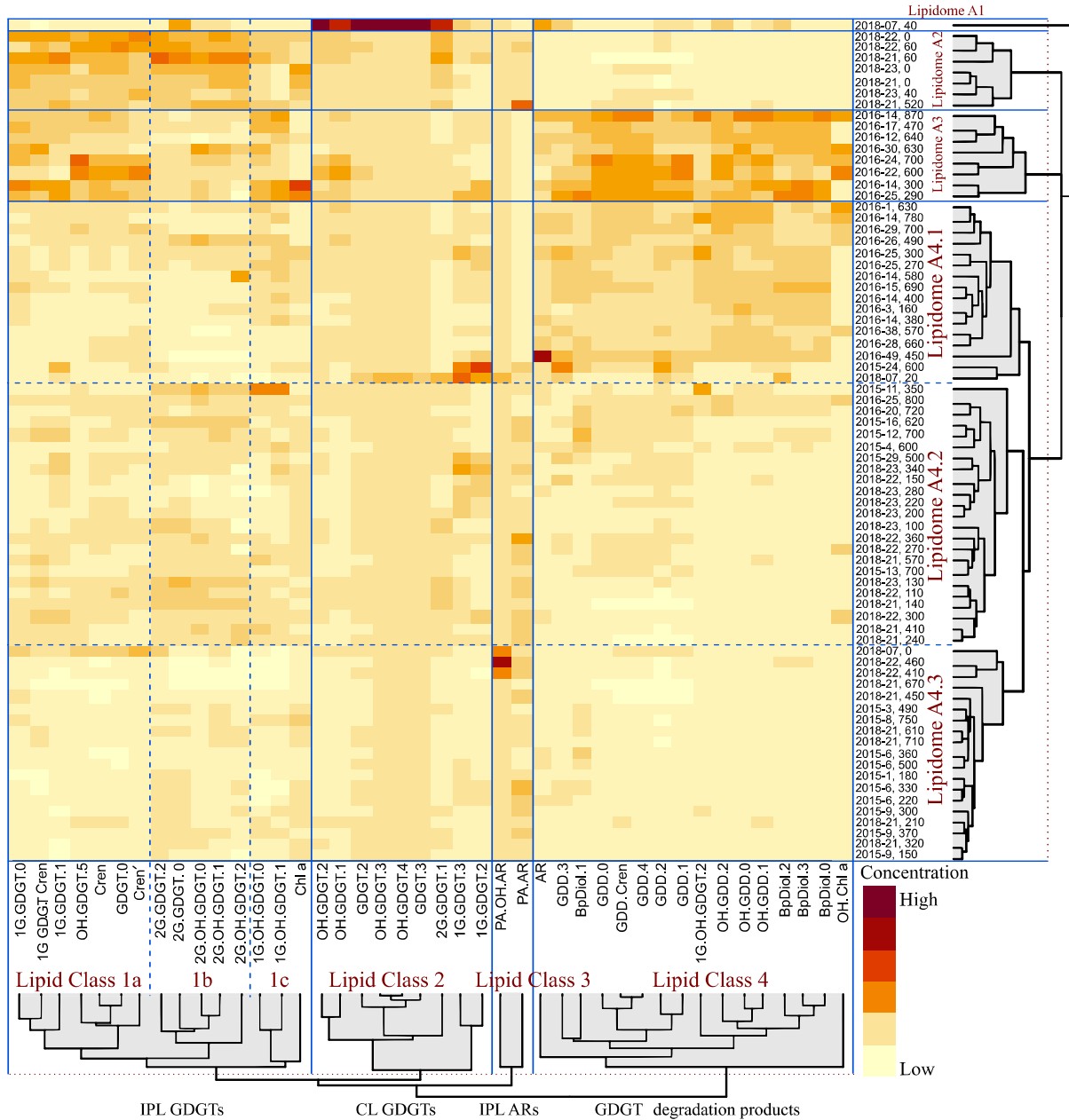

**Figure 8: HCA heatmap dendrogram generated displaying the z-scored concentration of archaeal lipid compounds in the 74 sediment samples. Rows mark the individual sediment samples. Columns display normalized z-score concentrations for the different lipid compounds (with the colour gradient indicating elevated concentrations with warmer colours). Red dotted lines indicate the similarity threshold used to distinguish clusters within the HCA dendrogram. Blue dotted lines distinguish sub-groups within clusters.**


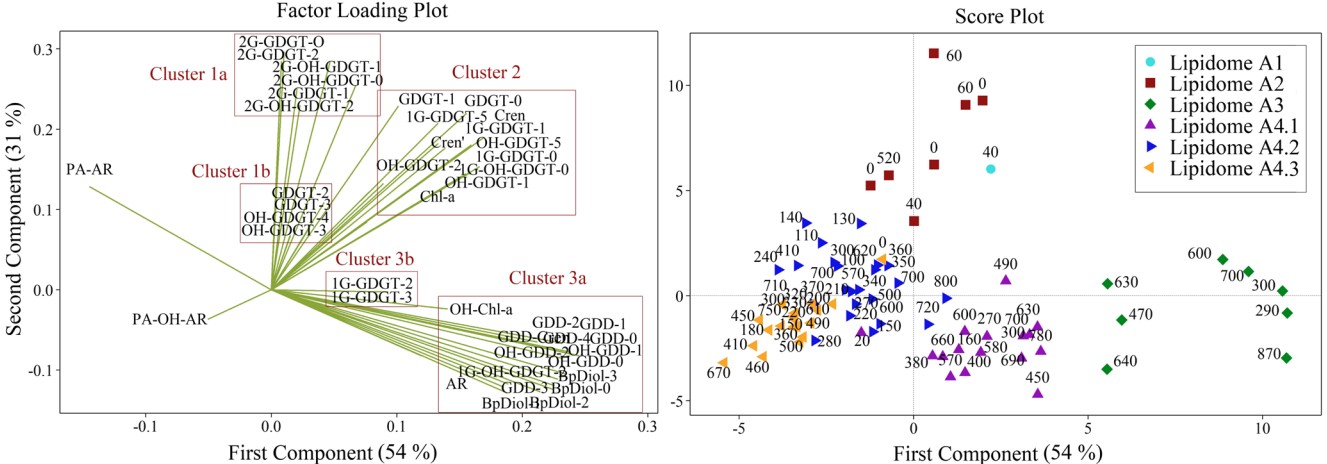

**Figure 9: Principal component analysis of archaeal lipid concentrations. Factor loadings plot (left) displaying five distinct clusters. Within the score plot (right), sediment samples are grouped corresponding to the six lipidomes (numerical labels indicate the sediment burial depth of each sample in centimetres).**

### 4.5.2 Microbial community reconstruction of lipidomes

The Scotian Slope lipidomes indicate systematic vertical change in archaeal microbial communities:

**Lipidome A1** is dominated by lipid class 2 compounds, including 1G-GDGT-2, GDGT-2, 1G-GDGT-3, GDGT-3, OH-GDGT-1 to -3 as well as 2G-GDGT-0 and AR (Fig. 8), which are commonly produced by methanotrophic archaea and Bathyarchaeia of cold seep environments (Blumenberg et al., 2004, Pancost et al., 2001; Zhang et al., 2011; Zhang et al., 2023). A high contribution of ANME-1 in this lipidome is shown by high concentrations of GDGT-2

and GDGT-3, in their IPL and CL forms. Thaumarchaeota group I.1a, is known to produce OH-GDGT compounds under cold, methane-rich conditions (Elling et al., 2017). Lipidome A1 includes samples whose core is associated with geochemical and microbial evidence of hydrocarbon seepage (Fowler et al., 2017).

**Lipidome A2** is marked by higher concentrations of lipid classes 1a, 1b, and 1c, which includes 1G- and 2G-GDGTs, as well as GDGT-0, Cren, and Cren′ that are known to be mainly synthesized by non-thermophilic ammonia-oxidizing

Thaumarchaeota inhabiting surface and shallow sediments (Schouten et al., 2000; 2002; 2013). A strong contribution from non-thermophilic Thaumarchaeota, particularly Thaumarchaeota group I.1a in lipidome A2 is supported by high concentrations of Cren and GDGT-0 (Schouten et al., 2000, 2002, 2013).

**Lipidomes A3 and A4s** are dominated by compounds from lipid class 4 that have higher contributions of GDGT degradation products such as GDDs, OH-GDDs, and Bp-Diols. Additionally, high concentration of Cren and GDGT-

0 supports a strong contribution from non-thermophilic Thaumarchaeota in this lipidome (Schouten et al., 2000; 2002; 2013). High AR and CL-DPs concentration in lipidomes A3 and A4.1 indicates increased contribution of methanogens and ANME groups (Pancost et al., 2011; Rossel et al., 2008) as well as methanogenic archaea (Liu et

al., 2012; Meador et al., 2014; Bauersachs et al., 2015). Lipidomes A4.2 and A4.3 exhibit the lowest concentrations of archaeal lipids including GDGT degradation products (lipid class 4). These lipidomes are likely dominated by Thaumarchaeota, particularly non-thermophilic groups such as Thaumarchaeota I.1a, which produce IPLs and core GDGTs (Elling et al., 2017).

Only one of the four stratigraphically controlled lipidomes (A1 from a prospective cold seep) is detectable by the surveyed environmental biomarker proxies (Fig. 7). This indicates biomarker proxies may not resolve some complex changes to archaeal community structures within marine sedimentary systems. Additionally, individual lipidomes do not typically produce correlated lipid pairs of diagenetic alteration steps (Section 4.4; Fig. 5).

## 4.6 Controlling factors and the spatial extent of the Scotian Slope lipostratigraphy

The six lipidomes have overlapping sediment depth ranges (Fig. 10). The spatial extent of each lipidome was interpolated from the outermost perimeter of depth equivalent core locations from which it could be resolved (Fig. 11) to reveal a slope wide archaeal lipostrigraphy. The mechanism for how these zones formed is unclear. One potential driver is sedimentology. Lithology affects porewater permeability, pH, and nutrient loading, which in turn could impact the archaeal community structure. To test this, sedimentological observations taken from cruise core logs (Table S2) were used to examine lipidomic linkages to lithology by statically comparing texture (suggestive of grain size changes), colour (corresponding to mineralogy and potentially redox changes), and sedimentary structures to lipidome occurrences (Fig. S7). These analyses did not produce meaningful statistical relationships. As such, sediment lithology (Fig. S7), changes in organic matter loading (Fig. 2), methane cycling (Fig. 7) and latitudinal position along the outlined slope quadrants (Fig. 1 and Fig. S6) do not fundamentally impact the archaeal lipostratigraphy (Fig. 11). Alternatively, microbial redox-controlled biogeochemical cycles are well defined within marine sedimentary systems. Slope-wide geochemical porewater profiles of sulfate and DIC (Fig. 2 and 10) indicate a complimentary occurrence of elevated microbial sulfate reduction. Lipidomes 3 and 4 occur near the sulphate alkalinity transition zone at ~3.25 mbsf (Fig. 10). This depth coincides with the beginning of the lipidomes A3 and A4.1. In this regard, sediment accumulation coupled to diffusion limitation appears to be driving stratified redox controlled biogeochemical zones that are mediated and affected by partially niche-partitioned microbial habitats. Additionally, burial depth was shown to affects lipid preservation (Figs 5 and 6).

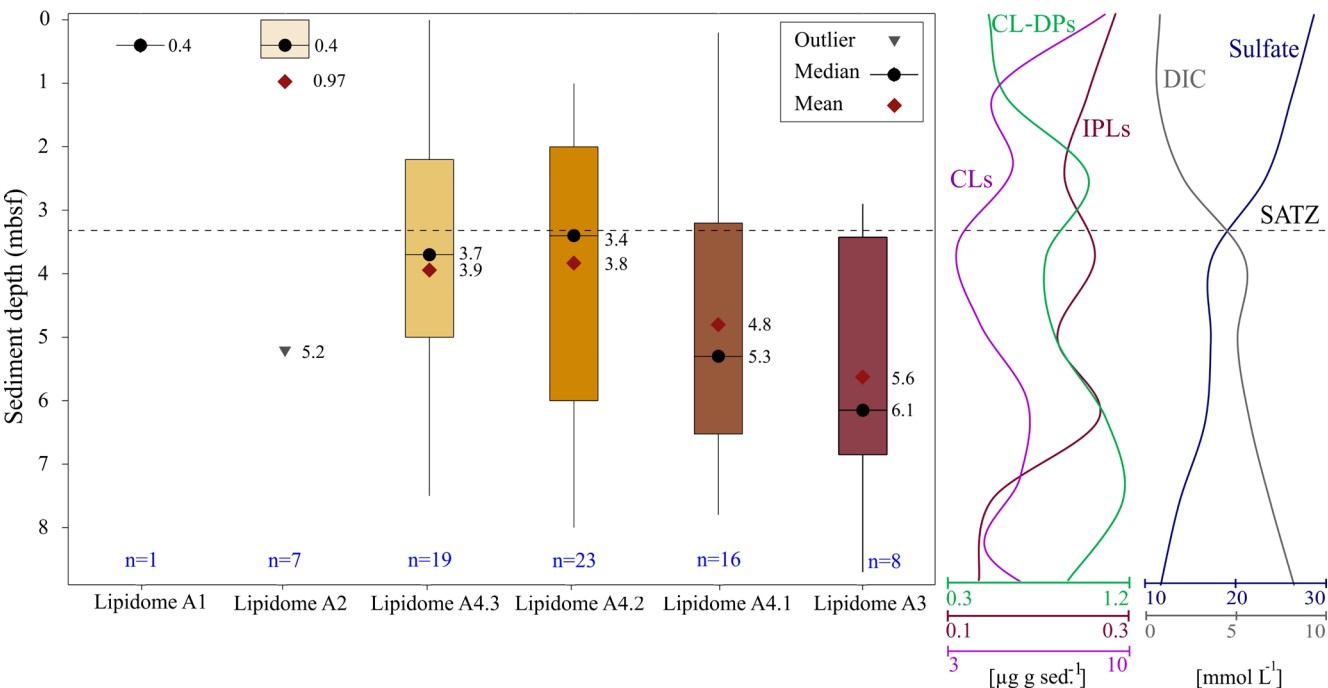

**Figure 10: Box-and-whisker plot displaying the interquartile sediment burial depth ranges of the six heatmap dendrogram resolved lipidomes (Fig. 8), along with the smoothed down core profile showing the slope averaged concentration of IPLs, CLs, CL-DPs, sulfate, and DIC made using the average concentration at 1 m depth intervals (Fig. 2). The SATZ is labelled at the cross-over of the two down core profiles. The number of sediment samples contributing to each lipidome in the box plot is reported in blue font. The values for outlier, mean, and median refers to one meter core depths.**

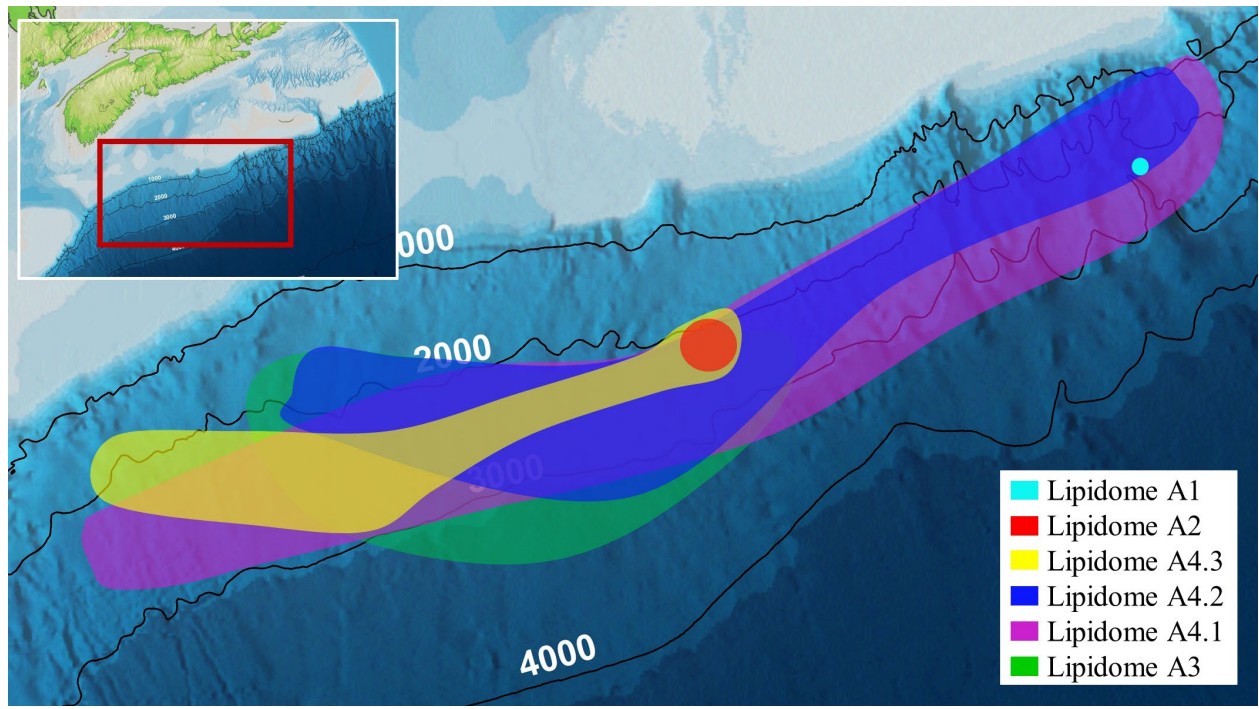

**Figure 11: Scotian Slope bathymetric map showing the spatial extent of the archaeal lipostratigraphy.**

## 5 Conclusions

This study quantified archaeal lipid classes from sediment collected along the Scotian Slope of Northeastern Atlantic, Canada. The survey extended to ~9 m below the ocean floor and spanned ~40,000 km$^2$, traversing ~3° of latitudinal change, with an overlying water column that increased from ~1500 to 3500 m depth. Archaeal lipids were found to be sourced from both the water column and ocean floor sediments. Measured diagenetic decay rates indicate the chemical structure of a lipid impacts its preservation with typically higher CL-DP and CL rates arising relative to that of IPLs. Additionally, lipid diversity and abundance systematically vary with sediment depth forming four stratigraphically overlapping lipidomes with high ANME-1 contributions. One of these represents contributions from an archaeal community impacted by an isolated seep environment. The other three mark ambient sediment lipidomes that reflect ~~27±4 Kyr of changing water column loading, mixed with in situ sedimentary additions from subsurface archaeal communities. These lipidomes are further augmented by increasing abundances of archaeal lipid degradation products. The ambient sediment lipidomes appear highly conserved across the latitudinal extent of the study area marking a resolvable shallow sediment lipostratigraphy for the Scotian Slope.

## Author contribution

NA and GTV designed the research and wrote the manuscript. NA generated the lipid data and provided the data analysis. UU provided technical support with lipid identification. NM provided geographical data. AM provided logistical support for sampling and project completion. ER and PG provided analytical support with bulk geochemistry and molecular analyses. MGF provided technical support for the interpretation of site localities. JNB provided technical support and helped to collect samples. VB provided technical support for porewater analysis.

## Competing interests

The authors declare that they have no conflict of interest.

## Acknowledgments

Thanks to the Nova Scotia Provincial Government for support of this project and spear-heading the various research cruises and surveys that made the integrative data collection possible. A special thanks is given to Carey Ryan of Net Zero Atlantic, who was the GAPP Partnership Program manager responsible for the larger group research initiatives. Carey also managed the procuration and organization of the research cruises. We would also like to acknowledge that Julius Lipp, Florence Schubotz, and Kai-Uwe Hinrichs of MARUM, University of Bremen were instrumental in helping to first develop SMU's techniques in lipidomics. An additional special thanks is given to Kim Doane, the executive director of the Subsurface and Offshore Energy Branch of the Department of Energy for her support of the Organic Geochemistry Research Group at Saint Mary's University. Sediment samples analysed in this study were collected during three marine field programs led by the Geological Survey of Canada under a joint collaborative research agreement with Nova Scotian Department of Natural Resources and Renewables. Kate Jarrett of Natural Resources Canada provided the core logs that were used to describe the sediment lithologies used for the study. Nikita Lakhanpal provided porewater analyses for this study.

## Financial support

Funding for this project was sourced from: Genome Atlantic and Genome Canada; Research Nova Scotia (grant no. 2142); Mitacs (grant no. IT12481 and IT29547), NetZero Atlantic and the Nova Scotia This research has been supported by the Natural Sciences and Engineering Research Council of Canada (grant no. RGPIN-2018-06147; NSERC), NSERC Canada Research Chairs (CRC) program, Canada Foundation for Innovation (CFI; JELF–CRC, John R. Evans Leaders Fund), NSERC Discovery Grants program (application no. RGPIN-2017-05822).

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
