# Peer review of "Archaeal lipostratigraphy of the Scotian Slope shallow sediments, Atlantic Canada"

_EGUsphere, 2025_

## Referee Comment (RC1)

**General Comments:**

The authors present a large, detailed lipidomic dataset focusing on archaeal membrane lipids and identifying distinct lipidomic patterns across an extensive area across the Scotian Slope. To my knowledge, this is the first comprehensive lipidomic study in the deep marine Scotian Slope system. The study encompasses an impressive amount of sediment cores (32) with great geographic coverage and spatial resolution. They used hierarchical cluster analysis and principal component analysis to identify distinct lipidomes that reflect the local archaeal communities, with one highlighting a cold seep area. In general, this study highlights great spatial consistency in archaeal lipidome patterns, suggesting that lipostratigraphy is conserved in this deep marine setting.

The authors demonstrate great care in lipid analysis, such as the normalization approach using a TOC decay model to correct for degradation to improve the quantitative comparability of lipid data. The dual normalization approach using lipid concentration based on extracted sediment and TOC mass is novel and well executed to strengthen the source assignment of specific lipids (sedimentary vs. upper water column).

The use of two multivariate statistical approaches (PCA and HCA) for the identification of reproducible lipidome clusters adds depth and statistical strength and is appropriate given the large number of samples. The agreement between the two methods strengthens the classification of distinct lipid assemblages.

I like the dual statistical approach, but I would argue that the description of the methodology of the HCA and PCA in 2.7 is a bit short. I am not an expert, but I think that there are likely parameters (clustering thresholds, linkage methods [Ward, Average …]) that could be mentioned here to ensure reproducibility.

In my opinion, the manuscript could generally be shortened/condensed. In some places, the sentences feel wordy or repetitive. However, in general, condensing the manuscript would improve the clarity for the reader. In addition, when revising the manuscript, more "transitional phrases" could be incorporated to improve the flow.

There are some grammar, language, and spelling issues or errors, such as missing hyphens, that I believe can easily be fixed. I mentioned a few things I noticed in the "Technical issues" section; however, my list is incomplete.

Some figures are a bit crowded. In a later section, I made some suggestions to improve the clarity of some individual figures.

Line 280: "Resolving archaeal lipid source inputs is difficult and the subject of long-term disagreement." I think this sentence is important, and in my opinion, the difficulties in source attribution solely based on lipids deserve a bit more attention or acknowledgment. If DNA-based evidence (e.g., 16S rRNA gene sequencing, metagenomics) for these samples is unavailable, it should be explicitly acknowledged in the discussion. In the Introduction, you mention that genomic analyses, including 16S rRNA amplicon sequencing and metagenomic profiling, have been done in the area, highlighting several papers. For example, ANME-1 has been found in comparable sites within the Scotian Slope in previous papers using genomic

evidence. You could further highlight this in the discussion and strengthen your lipid source attributions. The Dong et al. 2020 paper seems very interesting but is only cited in the Introduction.

In general, the authors put much emphasis on ANME and Thaumarchaeota, which is understandable given the lipids and proxies used. However, it would be great to acknowledge that marine sediment likely harbors other archaea as well (e.g., Bathyarchaea), which also contribute to the archaeal lipid pool.

$\delta 13C$ values of lipids (e.g., archaeol) for further evidence of ANME-2-mediated methanotrophy could also help strengthen source attributions; however, I acknowledge that this analysis might not be possible and is beyond the scope of the manuscript.

**Abstract:**

The abstract currently has > 430 words and has a high density of information. I would consider shortening it to a maximum of 300 words. See examples below:

In lines 27–28, the phrases "which are largely sourced from living cells" or "…that collectively are sourced from different alteration stages following the death of the cell" are unnecessary in an abstract.

The description of the Scotian Slope at the very beginning could also be condensed. Some of the precise information, like the depth gradient from 400 to 5 km and the maximum sediment thickness of 24 km, could be saved for the site description in the methods or the Introduction.

Line 22–23: Precise numbers such as "40,000 km²" or "3° of latitudinal change" could be mentioned in the method section instead to condense the abstract.

**Introduction:**

Line 43: Spelling error in hydrocarbon seep

Line 54: "dominant" instead of "dominate"

Line 57: "Indicator of" instead of "indicator to"

Line 73: Rephrasing "….lipids are formed from isoprenoidal…." to "…lipids consist of isoprenoidal…"

Line 75: Be careful with the phrasing. It sounds like only glycolipids and phospholipids exist; however, IPLs are much more diverse. For example, IPLs with amino-based head groups, such as ornithine lipids, are also quite common.

Line 75–78: There is a repeated definition of CL as degraded remnants of IPLs. That can be condensed.

Line 81–82: Consider rephrasing the sentence. "Significant" seems a bit vague in this sentence and could be interpreted in different ways. Making the connection to the following sentence clearer would help.

Line 83: "such as" instead of "like"

Line 84: Consider adding specific environmental conditions that can be reconstructed as examples. I think the two references focus only on sea-surface temperatures.

Line 89: Add an "and" between "geochemical, archaeal lipid proxy ratios"

**Methods:**

Line 106: Consider specifying the "additional geochemical evidence" that indicated hydrocarbon seepage

Table 1: The font size is quite small. Maybe consider putting this table in the supplements. It is a bit overwhelming. Alternatively, you could put the full table into the supplement but add a condensed version of the most important information here. I think that information like the coordinates, top and bottom core sample depth, and extracted sediment mass are better fit for the supplements.

Line 157: Spelling mistake in "1-alkyl-2-acetoyl-sn-glycero-3-phosphocholine" should be "1-alkyl-2-acetyl-sn-glycero-3-phosphocholine"

Line 159: Consider rephrasing "following additional protocols" to "following adaptations from Bentley et al. (2021)" or "following modifications from Bentley et al. (2021)." Otherwise, it sounds a bit like you are using additional extraction protocols other than the Bligh and Dyer protocol.

Line 161: Consider removing the phrase "living".

**Results:**

Line 225: You mentioned "plant-based" Chlorophyll a; however, could algae and cyanobacteria not also be contributing sources for chlorophyll?

Line 228: Consider rephrasing "0–3" to "GDGT-0 to GDGT-3" or "1G-GDGTs comprising 0–3 rings" for clarity.

Line 228–232: Glycerol dialkyl glycerol tetraether is spelled out multiple times. For better readability, consider using the abbreviation. For example, instead of "Hydroxyl diglycosidic glycerol dialkyl glycerol tetraethers," write "Hydroxylated 2G-GDGT" since 2G-GDGT was already defined before.

**Discussion:**

Line 257–260: This section seems somewhat redundant. The first sentence already states that isoprenoidal GDGTs are mainly produced by ammonia-oxidizing Thaumarchaeota, while the second sentence essentially repeats this, referring to specific GDGT distributions. Consider rephrasing or condensing this part. Also, to my knowledge, there are currently no confirmed thermophilic Thaumarchaeota, so the term "non-thermophilic" may be unnecessary.

Line 270: The phrase "methanogenic microorganism" may be misleading, as archaeol is generally produced exclusively by archaea, not other microorganisms.

Lines 273–275: These sentences may be more appropriate for the Results section.

Line 308: "GDGT moieties" instead of "GDGTs moieties."

Line 310: Consider rephrasing to "…distribution patterns of intact lipids from presumably viable sedimentary archaea differ significantly from …." or something similar.

Line 317: Consider using "reflect" instead of "arise from"

Line 318: A spelling error in "six group(s) of samples." Also, consider maintaining consistency in how numbers are presented. Sometimes they are written out (e.g., "six"), while other times they are shown as numerals (e.g., "6").

Line 322: Remove "analysis" as PCA already includes "analysis"

Line 393: Spelling error in ANME

Line 393: You mention that the increasing IPL-MI increases with depth, highlighting a higher proportion of ANME-1 and 2. However, I believe ANME-2 are not thought to be producers of GDGTs and instead produce diether lipids. Thus, I do not think mentioning ANME-2 specifically in this sentence is correct.

**Conclusion:**

Line 441: I think the suggestion that the increase in degradation products resulting from heterotrophy by microbial sulfate-reducing bacteria may be overly speculative for the conclusion as there is no direct evidence for this process.

Line 441 to 442: This sentence is also used in the abstract. Maybe consider rephrasing it slightly here to avoid exact repetition, especially since it is one of the main findings.

Line 443–445: The very end of the conclusion ends with you highlighting a limitation of this study and mentioning that these biomarker proxies did not fully resolve the existing biostratigraphy. While it is important to inform readers of limitations and room for future development, I would not put this as your closing statement. Maybe reorder the conclusion and end the manuscript with a strong statement emphasizing a key finding or novel aspect of your study.

**Figures in general:**

The font size should be increased for better readability, especially in Fig. 3, 4, and 8.

**Figure 3:**

Consider increasing the font size of the legend and the x-axis. While it may still be readable, I think it is overly small. I like that this figure gives the reader a great overview. All discussed lipids and the corresponding depth profiles are visualized in one image. However, it also looks a bit crowded. I want to suggest some modifications that may improve the figure:

1.	Consider making it a bit more obvious where one quadrant starts and another begins, perhaps by highlighting quadrants B and D in light grey or another color of your preference.

2.	In general, you could change the absolute abundance column to a line plot and make the column narrower to save space. Effectively, the relative abundances are shown twice. Once in the relative abundance plots and again in the bar plot, including absolute abundance. I think that might be redundant. This is particularly true for the photosynthetic pigments, where I recommend displaying only the absolute abundance plot.

**Figures 6 and 7** may deserve more attention as they are somewhat underexplained. I assume they primarily serve to confirm the results shown in Figure 5, but providing more guidance for the reader when discussing these figures could be helpful.

In the caption of Figure 6, the SATZ is mentioned. Including a visual marker of this boundary would be helpful, as it appears to be a central geochemical transition. This would make it easier for readers to connect the SATZ to the lipidome shifts.

**Figure 9:**

If I did not miss it, you never explicitly acknowledge Figure 9 in the text. It should be put into the supplements, or you should acknowledge it in the text somewhere. If possible, I would include the core location in the figure. Also, it is unclear how exactly the boundaries of the specific lipidomes were determined. I assume it is interpolated based on the core location; thus, it would help to see them in the image. Since this image shows one of the main conclusions of the manuscript, I would elaborate a bit more on how it was created.

**Supplements:**

Table S2: In the references, you write Stuart et al., 2004. I assume you mean Sturt et al., 2004.

---

## Author Comment (AC2)

**Response to Reviewer 2**

We thank Reviewer 2 for a thoughtful critique with additional insights that further capture issues addressed in Reviewer 1's comments. A revisited literature review is on the way that includes the listed references as well as additional papers publicly available to us. Review 2 also highlights three points changes that need to be addressed in the rewrite. These points include:

1. Revision to the concept of IPL and CL as proxies for living and fossil lipids that have a predictive diagenetic pathway.
2. The use of quantitative mass spectral measurements of chlorophylls for lipid sourcing assignments.
3. The interpretation of glycerol dialkyl diethers (GDDs) as degradation products.

Point 1. We realize that there is a gap in the paper's use of glycosidic-based IPLs as indicators of living and recently deceased cells as well as indicating core lipids are the older hydrolyzed remains of IPLs. This will have to be reworked in the paper. We do, however, find this to be an optimal opportunity to further test the hypothesis within our data set. As such, we will be adding the below two figures to the text. Figure 1 further establishes lipid sourcing relationships under the pretext of a simplified, but expected, diagenetic alteration pathway. The Scotian Slope samples are grouped by their resolved depth-dependent lipidomes as well as marking a single sample set. Collectively these data points to genetic affinities across lipidomes for the coupled transitions:1G-GDGTs to GDGTs ($R^2 = 0.48$), GDGTs to GDDs ($R^2 = 0.59$; only for the continuous series of lipidomes A3 and A4s, which represent a deeper sediment interval), and GDD to BpDiol ($R^2 = 0.75$).

Figure 2 provides additional evidence for the diagenetic alteration of specific biomarker classes over the surveyed sediment depths. Specifically, 2G- and 1G-AR appear to systematically degrade to AR. Likewise, GDDs also appear to give rise to BpDiols. 2G-GDGTs and 2G-OH-GDGTs have very slight decreases relative to their CL equivalents. Not following the anticipated alteration pathway are 1G-GDGTs and 1G-OH-GDGTs. These compounds appear to be quite resilient within the sedimentary environment over the surveyed depth range. Based on these outcomes, we will be modifying the text to indicate that diagenetic changes do proceed in a limited fashion with respect to expected alteration pathways for some, but not all, identified compound classes.

[Figure]

Figure 1. Cross plots with PLS regressions for the concentrations of various lipid classes associated within an expected diagenetic alteration pathway (bottom flow chart).

[Figure]

Figure 2. Lipid precursor to product pair changes with core depth.

Point 2. Chlorophyll extraction requires a highly polar solvent like methanol or acetone. For the modified Bligh and Dyer method, the final two steps use ~20 ml MeOH/DCM [5:1; v/v] mix, which is sufficient to solubilize these pigments, where they can be easily separated using HPLC techniques. The detection of chlorophyll can be done in various ways. While photospectrometric techniques are quite common, there is also a large literature that has used mass spectral methods for chlorophyll measurements (e.g., Chen et al., 2015; Milenkovic et al., 2012). Nonetheless, we were quite concerned with the comment and did an expensive audit of our use of these pigments across all other projects where these biomarkers are detected. In all cases, we see expected patterns within sample sets.

Water column surveys invariably record high concentrations (~8 ug·L$^{-1}$ or 8 mg·m$^{-3}$ – papers in prep.) in samples collected from the euphotic zone. This is close to reported annual amplitude of surface chlorophyll concentration data obtained from the SeaWiFS satellite that report Scotian Slope values that range from 3–10 mg·m$^{-3}$ (Fennel, 2010). In deeper water column samples these compounds are either at very low concentrations or are absent. Ambient ocean floor sediments have higher loadings than cold seep sediments that are heavily populated by macrofauna.

For our study, we also show an expected pattern of altered pigment (hydroxychlorophyll) with increasing sediment depth. Given all samples are randomized at each processing step, the probability of this trend being an artifact is quite low. While we do not have the opportunity at this juncture to test the extraction efficiency relative to alternative techniques used by other groups to determine potential offsets in the absolute quantitative measures of these compounds, we can say our precision and ability to measure firstorder changes within environmental sample sets is robust for its use in validating the source of archaeal lipids.

Point 3. IPL-based GDDs were not found in our study with the methods used for lipid isolation and detection. We further find a high statistical relationship between GDDs and BpDiols (Figure 1) as well as a pronounced down core change in the proportion of GDDs to BpDiols (Figure 2) to be strong evidence that these compounds are early-stage degradation products of GDGTs. This relationship is further bolstered by 1G-GDGTs being correlated to GDGTs. GDGTs are only correlated with GDDs for the deeper depth continuous lipidomes A3 to A4 (Figure 1). A down core decrease in the relative abundance of GDGTs to GDDs is further observed in Figure 2. That said, we should have indicated that alternative sources for these compounds may exist and cited your paper.

Apart from these three points, all provided references will be carefully reviewed and added where appropriate to the revised manuscript. We will also take time to carefully edit the text (especially section 4.1) to improve the interpretation and readability of the paper, which should also shorten the length of the text to some degree.

**References**

Chen, Kewei, José Julián Ríos, María Roca, and Antonio Pérez-Gálvez. "Development of an Accurate and High-Throughput Methodology for Structural Comprehension of Chlorophylls Derivatives. (II) Dephytylated Derivatives." *Journal of Chromatography A* 1412 (September 2015): 90–99. https://doi.org/10.1016/j.chroma.2015.08.007.

Fennel, K. 2010. The role of continental shelves in nitrogen and carbon cycling. Ocean Sci. Discuss., 7, 177–205.

Milenković, Sanja M, Jelena B Zvezdanović, Tatjana D Anđelković, and Dejan Z Marković. "The Identification of Chlorophyll and Its Derivatives in the Pigment Mixtures: HPLC-Chromatography, Visible and Mass Spectroscopy Studies." *Advanced Technologies*, 2012.

Reviewer 1

General Comments:

The authors present a large, detailed lipidomic dataset focusing on archaeal membrane lipids and identifying distinct lipidomic patterns across an extensive area across the Scotian Slope. To my knowledge, this is the first comprehensive lipidomic study in the deep marine Scotian Slope system. The study encompasses an impressive amount of sediment cores (32) with great geographic coverage and spatial resolution. They used hierarchical cluster analysis and principal component analysis to identify distinct lipidomes that reflect the local archaeal communities, with one highlighting a cold seep area. In general, this study highlights great spatial consistency in archaeal lipidome patterns, suggesting that lipostratigraphy is conserved in this deep marine setting.

The authors demonstrate great care in lipid analysis, such as the normalization approach using a TOC decay model to correct for degradation to improve the quantitative comparability of lipid data. The dual normalization approach using lipid concentration based on extracted sediment and TOC mass is novel and well executed to strengthen the source assignment of specific lipids (sedimentary vs. upper water column).

The use of two multivariate statistical approaches (PCA and HCA) for the identification of reproducible lipidome clusters adds depth and statistical strength and is appropriate given the large number of samples. The agreement between the two methods strengthens the classification of distinct lipid assemblages.

I like the dual statistical approach, but I would argue that the description of the methodology of the HCA and PCA in 2.7 is a bit short. I am not an expert, but I think that there are likely parameters (clustering thresholds, linkage methods [Ward, Average …]) that could be mentioned here to ensure reproducibility.

In my opinion, the manuscript could generally be shortened/condensed. In some places, the sentences feel wordy or repetitive. However, in general, condensing the manuscript would improve the clarity for the reader. In addition, when revising the manuscript, more "transitional phrases" could be incorporated to improve the flow.

There are some grammar, language, and spelling issues or errors, such as missing hyphens, that I believe can easily be fixed. I mentioned a few things I noticed in the "Technical issues" section; however, my list is incomplete.

Some figures are a bit crowded. In a later section, I made some suggestions to improve the clarity of some individual figures.

Line 280: "Resolving archaeal lipid source inputs is difficult and the subject of long-term disagreement." I think this sentence is important, and in my opinion, the difficulties in source attribution solely based on lipids deserve a bit more attention or acknowledgment. If DNA-based evidence (e.g., 16S rRNA gene sequencing, metagenomics) for these samples is unavailable, it should be explicitly acknowledged in the discussion. In the Introduction, you mention that genomic analyses, including 16S rRNA amplicon sequencing and metagenomic profiling, have been done in the area, highlighting several papers. For example, ANME-1 has been found in comparable sites within the Scotian Slope in previous papers using genomic evidence. You could further highlight this in the discussion and strengthen your lipid source

attributions. The Dong et al. 2020 paper seems very interesting but is only cited in the Introduction.

In general, the authors put much emphasis on ANME and Thaumarchaeota, which is understandable given the lipids and proxies used. However, it would be great to acknowledge that marine sediment likely harbors other archaea as well (e.g., Bathyarchaea), which also contribute to the archaeal lipid pool.

$\delta$13C values of lipids (e.g., archaeol) for further evidence of ANME-2-mediated methanotrophy could also help strengthen source attributions; however, I acknowledge that this analysis might not be possible and is beyond the scope of the manuscript.

Abstract:

The abstract currently has > 430 words and has a high density of information. I would consider shortening it to a maximum of 300 words. See examples below:

In lines 27–28, the phrases "which are largely sourced from living cells" or "…that collectively are sourced from different alteration stages following the death of the cell" are unnecessary in an abstract.

The description of the Scotian Slope at the very beginning could also be condensed. Some of the precise information, like the depth gradient from 400 to 5 km and the maximum sediment thickness of 24 km, could be saved for the site description in the methods or the Introduction.

Line 22–23: Precise numbers such as "40,000 km²" or "3° of latitudinal change" could be mentioned in the method section instead to condense the abstract.

Introduction:

Line 43: Spelling error in hydrocarbon seep

Line 54: "dominant" instead of "dominate"

Line 57: "Indicator of" instead of "indicator to"

Line 73: Rephrasing "….lipids are formed from isoprenoidal…." to "…lipids consist of isoprenoidal…"

Line 75: Be careful with the phrasing. It sounds like only glycolipids and phospholipids exist; however, IPLs are much more diverse. For example, IPLs with amino-based head groups, such as ornithine lipids, are also quite common.

Line 75–78: There is a repeated definition of CL as degraded remnants of IPLs. That can be condensed.

Line 81–82: Consider rephrasing the sentence. "Significant" seems a bit vague in this sentence and could be interpreted in different ways. Making the connection to the following sentence clearer would help.

Line 83: "such as" instead of "like"

Line 84: Consider adding specific environmental conditions that can be reconstructed as examples. I think the two references focus only on sea-surface temperatures.

Line 89: Add an "and" between "geochemical, archaeal lipid proxy ratios"

Methods:

Line 106: Consider specifying the "additional geochemical evidence" that indicated hydrocarbon seepage

Table 1: The font size is quite small. Maybe consider putting this table in the supplements. It is a bit overwhelming. Alternatively, you could put the full table into the supplement but add a condensed version of the most important information here. I think that information like the coordinates, top and bottom core sample depth, and extracted sediment mass are better fit for the supplements.

Line 157: Spelling mistake in "1-alkyl-2-acetoyl-sn-glycero-3-phosphocholine" should be "1-alkyl-2-acetyl-sn-glycero-3-phosphocholine"
Line 159: Consider rephrasing "following additional protocols" to "following adaptations from Bentley et al. (2021)" or "following modifications from Bentley et al. (2021)." Otherwise, it sounds a bit like you are using additional extraction protocols other than the Bligh and Dyer protocol.

Line 161: Consider removing the phrase "living".
Results:

Line 225: You mentioned "plant-based" Chlorophyll a; however, could algae and cyanobacteria not also be contributing sources for chlorophyll?

Line 228: Consider rephrasing "0–3" to "GDGT-0 to GDGT-3" or "1G-GDGTs comprising 0–3 rings" for clarity.

Line 228–232: Glycerol dialkyl glycerol tetraether is spelled out multiple times. For better readability, consider using the abbreviation. For example, instead of "Hydroxyl diglycosidic glycerol dialkyl glycerol tetraethers," write "Hydroxylated 2G-GDGT" since 2G-GDGT was already defined before.

Discussion:

Line 257–260: This section seems somewhat redundant. The first sentence already states that isoprenoidal GDGTs are mainly produced by ammonia-oxidizing Thaumarchaeota, while the second sentence essentially repeats this, referring to specific GDGT distributions. Consider rephrasing or condensing this part. Also, to my knowledge, there are currently no confirmed thermophilic Thaumarchaeota, so the term "non-thermophilic" may be unnecessary.

Line 270: The phrase "methanogenic microorganism" may be misleading, as archaeol is generally produced exclusively by archaea, not other microorganisms.

Lines 273–275: These sentences may be more appropriate for the Results section.

Line 308: "GDGT moieties" instead of "GDGTs moieties."

Line 310: Consider rephrasing to "…distribution patterns of intact lipids from presumably viable sedimentary archaea differ significantly from …." or something similar.

Line 317: Consider using "reflect" instead of "arise from"

Line 318: A spelling error in "six group(s) of samples." Also, consider maintaining consistency in how numbers are presented. Sometimes they are written out (e.g., "six"), while other times they are shown as numerals (e.g., "6").

Line 322: Remove "analysis" as PCA already includes "analysis"

Line 393: Spelling error in ANME

Line 393: You mention that the increasing IPL-MI increases with depth, highlighting a higher proportion of ANME-1 and 2. However, I believe ANME-2 are not thought to be producers of GDGTs and instead produce diether lipids. Thus, I do not think mentioning ANME-2 specifically in this sentence is correct.
Conclusion:

Line 441: I think the suggestion that the increase in degradation products resulting from heterotrophy by microbial sulfate-reducing bacteria may be overly speculative for the conclusion as there is no direct evidence for this process.

Line 441 to 442: This sentence is also used in the abstract. Maybe consider rephrasing it slightly here to avoid exact repetition, especially since it is one of the main findings.

Line 443–445: The very end of the conclusion ends with you highlighting a limitation of this study and mentioning that these biomarker proxies did not fully resolve the existing biostratigraphy. While it is important to inform readers of limitations and room for future development, I would not put this as your closing statement. Maybe reorder the conclusion and end the manuscript with a strong statement emphasizing a key finding or novel aspect of your study.

Figures in general:

The font size should be increased for better readability, especially in Fig. 3, 4, and 8.

Figure 3:

Consider increasing the font size of the legend and the x-axis. While it may still be readable, I think it is overly small. I like that this figure gives the reader a great overview. All discussed lipids and the corresponding depth profiles are visualized in one image. However, it also looks a bit crowded. I want to suggest some modifications that may improve the figure:

1. Consider making it a bit more obvious where one quadrant starts and another begins, perhaps by highlighting quadrants B and D in light grey or another color of your preference.

2. In general, you could change the absolute abundance column to a line plot and make the

column narrower to save space. Effectively, the relative abundances are shown twice. Once in the relative abundance plots and again in the bar plot, including absolute abundance. I think that might be redundant. This is particularly true for the photosynthetic pigments, where I recommend displaying only the absolute abundance plot.

Figures 6 and 7 may deserve more attention as they are somewhat underexplained. I assume they primarily serve to confirm the results shown in Figure 5, but providing more guidance for the reader when discussing these figures could be helpful.

In the caption of Figure 6, the SATZ is mentioned. Including a visual marker of this boundary would be helpful, as it appears to be a central geochemical transition. This would make it easier for readers to connect the SATZ to the lipidome shifts.

Figure 9:

If I did not miss it, you never explicitly acknowledge Figure 9 in the text. It should be put into the supplements, or you should acknowledge it in the text somewhere. If possible, I would include the core location in the figure. Also, it is unclear how exactly the boundaries of the specific lipidomes were determined. I assume it is interpolated based on the core location; thus, it would help to see them in the image. Since this image shows one of the main conclusions of the manuscript, I would elaborate a bit more on how it was created.

Supplements:

Table S2: In the references, you write Stuart et al., 2004. I assume you mean Sturt et al., 2004.

**Reviewer 2**

In the manuscript "Archaeal lipostratigraphy of the Scotian Slope shallow sediments, Atlantic Canada" Ahangarian et al. present the archaeal lipidomes in a wide range of sediments taken across the Scotian Slope. I was impressed by the large dataset which covers such a broad geographical range. I am generally accepting of the methods utilized and the lipid data itself looks interesting and valid. However, my concern about the manuscript is in the interpretation and discussion of the data. I feel that authors have not sufficiently reviewed the literature on archaea and archaeal lipids, which in turn has had a considerable effect on how they interpret their data.

One major issue is that the authors have made the assumption that glycosidic GDGTs are indicative of living archaea. This has been shown extensively to not be true, both experimentally and in the environment. Glycosidic GDGTs are known to be preserved for thousands and even millions of years. GDGTs with phospho-bound head groups are more accurate markers of living or recently living cells. However, the authors do not detect phospho-bound GDGTs, only some phospho-bound archaeols. The authors have failed to correctly interpret some of the literature they cite, such as the review of Schouten et al. (2013) which clearly states that "these IPLs may be of fossil origin, as degradation of ether lipids with a sugar head group, or even IPL GDGTs in general can proceed much more slowly than that of regular bacterial ester-bound IPLs.". I draw the authors' attention to more recent work that also shows that glycosidic IPL GDGTs cannot be used as markers for living archaea (Wu et al., 2019) and ask that the reviewers read this and the literature cited in its introduction. As a result of this error, amongst others listed below, the authors should go back and fully rewrite their manuscript to take this into account,

carefully reexamining their assertions from their data about "living, fossil and degraded lipids". Indeed, it would seem possible (or even probable) that the vast majority of the GDGTs detected originate from the overlying water column, not from the sediment itself. This is going to have considerable effect on the manuscript and hence I suggest major revisions to accomplish this.

Overall, I find it quite risky that the authors using the profiles of the photosynthetic pigment chl-a and its alteration product OH-chl-a in order to interpret the lipid data. The lipid extraction method used is not designed for pigment extraction, nor is the analysis method. As this pigment is notoriously easy to destroy with light, heat or acid, there is a risk that the depth profile of these compounds is an effect of variable extraction efficiency. The authors need to examine experimentally how rigorous their extraction methodology is for chlorophyll. As I stated above, OH-chl-a is an alteration product of chl-a (often formed during extraction), so I would not describe it as a photosynthetic pigment (Steele et al., 2018).

Another issue in terms of the authors' interpretation of the literature, is the assignment of GDDs as GDGT degradation products. This is indeed the conclusion of Hingley et al. (2024), however the other article cited, (Mitrović et al., 2023), makes a quite different statement "it seems unlikely that GDDs are a direct diagenetic product of GDGTs. Overall, the observations from our study support the theory of a joint cellular origin for GDGTs and GDDs.". It is ok if the authors of this manuscript to not agree with Mitrović et al. (2023), but they should not incorrectly present the conclusions of that study.

The authors also need to address the language used throughout. During revision this manuscript should be carefully read through and checked. As I have asked for major revisions, I will not do a line-by-line list of edits, as so much is likely to change. However here are some edits I would draw to the authors' attention.

Writing general – when rewriting this should be tightened up with fewer words and sharper sentences.

Line 21 and in more places throughout the manuscript (e.g. line 64). It's not clear here what the authors mean here by deep. Is that deep sea or deep sediment.

Line 53. Thaumarchaeota were reclassed a few years ago as Nitrososphaerota (Rinke et al., 2021). I'm not suggesting that you use this newer term throughout the manuscript, but it should be mentioned here.

Line 74. I think you should redefine all abbreviations in the introduction and not rely on them being defined in the abstract.– e.g. IPL and CL

Table 1 – this needs to be improved visually and a lot of the information (such as sediment extracted and TLE weight) moved to a supplementary table.

Figure 1 (and others) – seems very fuzzy.

Line 236. This isomer of crenarchaeol is not a regioisomer and is hence no longer called this (cf. Sinninghe Damsté et al., 2018). Please read the up-to-date literature on this subject.

Entire section 4.1. Many of lines read poorly and should be rewritten for clarity. Additionally, the authors emit some significant recent work on the GDGTs of Thaumarchaeota, e.g. Elling et al. (2017) and Bale et al. (2019) and on hydroxy GDGTs e.g. (Varma et al., 2024).

---

## Author Response (AR3)

**Biogeosciences Review**

We again thank the reviewers for the time and effort in improving our manuscript. Comments and a detailed account of the changes that have been made to the revised submission are presented below.

1.    I thank the authors for clarifying how TOC concentrations in each sample were corrected to an expected value of the original TOC based on the "regionalized Scotian Slope sediment TOC depth profile". I re-read the methods and still see the regionalized profile to be a key part of the way normalized lipid concentrations are calculated but also found that the method used to create this regionalized profile is not explained anywhere. Adding this explanation to the methods is needed.

I searched through the manuscript for evidence of how this regionalized profile was determined and the only clue is in the legend of the left-hand panel of figure 2 (no panel labels), a green line marked "Regression". It is not easy to tell what kind of regression this is except that it is clearly not linear. And again, I note that the fit is poor between the regression and data, overestimating the TOC concentration for almost all actual samples at shallow burial depths and underestimating it at deeper sediment depths. This poor fit would result in systematic error with a depth-based bias when calculating lipid concentrations normalized to the "original" TOC concentration considering how the regression model deviates from your data.

There are established models for organic matter degradation and TOC concentrations in sediment profiles like the classic model proposed by Middelburg, 1989 or the version that includes methane cycling proposed by Wallmann et al, 2006. The review by Arndt et al., 2013 contains a summary of models (Fig. 11) that includes a couple of published models made very close to the study location. Although a suggestion to go back to the data analysis stage is not going to be particularly welcome during a manuscript review, changing the regionalized TOC depth profile to one that is generated by a better, published model should not take too much work but would significantly improve on the robustness and accuracy of the normalized lipid concentrations and comparisons between them.

Recommend adding panel letters to Figure 2. This has been done.

References:
Middelburg, J. J. 1989. "A Simple Rate Model for Organic Matter Decomposition in Marine Sediments." Geochimica et Cosmochimica Acta 53 (7): 1577–81.

Wallmann, K., Aloisi, G., Haeckel, M., Obzhirov, A., Pavlova, G., Tishchenko, P., 2006. Kinetics of organic matter degradation, microbial methane generation and gas hydrate formation in anoxic marine sediments. Geochimica et Cosmochimica Acta 70, 3905–3927.

Arndt, S. , B. B. Jørgensen, D. E. LaRowe, J. J. Middelburg, R. D. Pancost, and P. Regnier. 2013. "Quantifying the Degradation of Organic Matter in Marine Sediments: A Review and Synthesis." Earth-Science Reviews 123 (August): 53–86.

In light of the above comments, we have reviewed the provided references and found that they do not conflict but support our technique. As such, these have been added to the paper in line 168. We have also rewritten this section for better clarity. The section heading has been changed to "Calculation of adjusted TOC concentrations for lipid normalization." We have added additional text to the beginning of the section which better clarifies the intent and direction of how sediment TOC was adjusted to factor out expected losses from diagenetic processes. We also realized that the prior regression model did induce bias to the lipids normalized at both shallow and deep intervals of the sampled sediment depth intervals. The logarithmic regression statistic was applied to 82 core sediment samples as the best fit line to the merged downcore TOC profile. The statistic produced an $R^2$ value of 0.48. We have examined the data analysis again and find the regression technique still produces the most optimal fit. However, in our reanalysis, we realized that two TOC samples deeper in the profile behave as outliers. We have removed those (now n=80). This produces a tighter fit to the data (particularly at the deeper sediment depths) with a new $R^2$ value of 0.58. The merged TOC normalized data (Fig. 4) was then renormalized to the lipid data, but adjusted values did not change the normalization profiles, and no further edits have therefore been made to those parts of the text.

2. I understand that the authors feel Discussion Section 4.1 is necessary to educate the readers of Biogeosciences but I would suggest this is highly unusual, unnecessary for most readers with an interest in this topic and makes a long paper even longer. I have never seen a discussion that has an entire section with no mention of the actual data at all; a discussion section exists to discuss the data. This material could be used to expand the short sections describing "Lipidome" A1-A4, although I do not think this is necessary. Those paragraphs are clear enough. It could also be turned into a mini review separate from this paper.

We realize there is considerable push-back on this point. Given the paper is long we have moved this section to the supplemental information and have included the below statement to mark the existence of this information for those who have additional interest in knowing lipid sourcing.

4.1 Chemotaxonomic relationships

Details on the chemotaxonomic relationships of the resolved Scotian Slope archaeal lipids can be found the Supplemental Information.

3. I also include responses to the final author comment copied here in quotes with my response below:

"While older uses of term 'lipidome' have been quite narrow such as being the full characterization of lipid molecular species and of their biological roles with respect to expression of proteins involved in lipid metabolism and function, including gene regulation or within a cell. More recent descriptions of the term loosely define it as the entire spectrum of lipids in a biological system (e.g. Seppaneo-Laakso and Oresic, 2009; Swinnen and Dehairs,

2022). That implicitly extends to ecological systems such that the adjective 'biological' is often dropped within environmental lipidomic based discussions. The term is therefore correctly applied in this study"

The authors' reply that their definition of "lipidome" (including both biological lipids and their degradation products), is different from many others that just include biological lipids including the references that were included. There is room for misunderstanding when implicitly extending this term to ecological systems when either dropping the "biological" or not dropping it are both done. Therefore, this use of the word should be defined at its first instance. For example at the end of the introduction on line 82, something like "distinct assemblages of archaeal lipids, referred to here as lipidomes"

The correction is done and the sentence now reads:

"The resolvable assemblages of archaeal lipids, referred to here as lipidomes, are examined across three spatial dimensions: sediment depth, distance down the continental slope, and along ~3° latitude change of the northwestern trend of the continental margin."

Also,

"We would also argue that one of the primary purposes of producing lipidomic studies is to establish reliable inferences on what comprises existing, past, and ancient microbiomes when other 'omics'-based approaches are either not available or are unsuited for use in in a particular study. The longevity of how well such interpretations hold up is of course an omnipresent issue impacting all organic geochemistry biomarker-based studies."

I will still maintain that attempting to infer archaeal community composition from lipid distributions when many important sedimentary bacteria (e.g. Bathyarchaeota) have uncharacterized lipid profiles pushes beyond the limits of current knowledge and is almost certainly inaccurate regardless of what we might wish lipids could tell us about existing, past, and ancient microbiomes. But, at this point, I'll leave that up to the editor and potentially the readers of this manuscript to judge the wisdom of doing this.

This is a truly valid statement. It is a piece of the puzzle. Additionally, while archaea and bacteria form commensal and symbiotic associations, we cannot detail these relationships here and the ability for lipids as a standalone proxy to do this is contingent on the acquisition of additional data (namely their compound specific isotope compositions). Nonetheless, we have additional manuscripts in prep that entirely focus on the bacterial lipid assemblages of the Scotian Slope and will be working to understand slope sediment community dynamics using these targets.

---

## Author Response (AR4)

Dear Dr. Naeher,

Thank you for your final evaluation of our paper. To complete the above requested change, we have re-written the beginning of section 4.5.2 Microbial community reconstruction of lipidomes (lines 416 to 419) to read:

"Although significant limits exist to the extent to which lipid distributions can resolve source organism taxonomy, they nonetheless still mark a quantitative measure of archaeal activity. In this study the resolve Scotian Slope lipidomes indicate systematic vertical change in archaeal microbial communities:"

Kind regards,

Todd Ventura